# Targeted mutagenesis of the herpesvirus fusogen central helix captures transition states

Momei Zhou [1] ✉, Benjamin Vollmer [2,3,4], Emily Machala [2,3,4], Muyuan Chen[5], Kay Grünewald [2,3,4], Ann M. Arvin[1,6,7], Wah Chiu [5,6,8] & Stefan L. Oliver [1]

Herpesviruses remain a burden for animal and human health, including the medically important varicella-zoster virus (VZV). Membrane fusion mediated by conserved core glycoproteins, the fusogen gB and the heterodimer gH-gL, enables herpesvirus cell entry. The ectodomain of gB orthologs has five domains and is proposed to transition from a prefusion to postfusion conformation but the functional relevance of the domains for this transition remains poorly defined. Here we describe structure-function studies of the VZV gB DIII central helix targeting residues [526]EHV[528]. Critically, a H527P mutation captures gB in a prefusion conformation as determined by cryo-EM, a loss of membrane fusion in a virus free assay, and failure of recombinant VZV to spread in cell monolayers. Importantly, two predominant cryo-EM structures of gB[H527P] are identified by 3D classification and focused refinement, suggesting they represented gB conformations in transition. These studies reveal gB DIII as a critical element for herpesvirus gB fusion function.

The *Herpesviridae* encompass pathogens that cause a wide range of medically and economically important diseases of humans and animals[1]. All herpesviruses are enveloped and use glycoproteins to initiate and drive membrane fusion between the virion and the host cell during entry[2]. The three glycoproteins, gB, gH, and gL, required for this process are conserved across the *Herpesviridae*[2,3]. Structure-function studies have revealed that the gB trimer is the primary fusogen and the gH-gL heterodimer has roles in cell tropism and gB regulation[1,4–10]. Incorporation of gB into the envelope of herpesvirus particles is complex. gB is transported to the plasma membrane of the infected cell, endocytosed, and trafficked to the trans-Golgi network for insertion into the herpesvirus envelope during morphogenesis[11]. To achieve fusion with cellular membranes, gB on the virion envelope is thought to undergo a conformational transition from a meta-stable prefusion structure to an energetically more stable postfusion state[2,12].

VZV, a member of the *Alphaherpesvirinae* and human host restricted pathogen, causes varicella (chickenpox), establishes latency in sensory ganglion neurons, and can reactivate to manifest as zoster (shingles)[13]. In addition to membrane fusion required for virion entry, gB-dependent cell-cell fusion is a hallmark of VZV pathogenesis[1]. Characteristic polykaryocytes form within infected tissues in vivo and are modeled in vitro by syncytia formation between MeWo cells[1]. VZV gB, together with gH-gL, are necessary and sufficient to induce cell-cell fusion in vitro[6,14–17]. Unlike its HSV gB orthologues, VZV gB is modified by furin cleavage post-translationally, which is needed for optimal skin lesion formation in vivo[14]. Critically, VZV induced fusion between differentiated cells has been linked to adverse health effects, including fusion of ganglion neurons and satellite cells associated with post-herpetic neuralgia (PHN), and fusion of endothelial cells in the brain vasculature linked to strokes[18,19].

[1]Department of Pediatrics, Stanford University School of Medicine, Stanford, CA, USA. [2]Centre for Structural Systems Biology (CSSB), Hamburg, Germany. [3]Department of Chemistry, University of Hamburg, Hamburg, Germany. [4]Leibniz Institute of Virology (LIV), Hamburg, Germany. [5]Division of Cryo-EM and Bioimaging SSRL, SLAC National Accelerator Laboratory, Menlo Park, CA 94025, USA. [6]Department of Microbiology & Immunology, Stanford University School of Medicine, Stanford, CA, USA. [7]Vir Biotechnology Inc, San Francisco, CA, USA. [8]Bioengineering, Stanford University School of Medicine, Stanford, CA 94305, USA. ✉e-mail: mzhou6@stanford.edu

Despite substantial amino acid sequence differences, the gB orthologues from alpha-, beta- and gammaherpesvirus subfamilies adopt remarkably similar structures in their postfusion ectodomain conformations, consisting of homotrimers with five domains, DI-DV[3,20–27]. This architecture led to the classification of gB orthologues as type III fusogens based on similarities to postfusion forms of vesicular stomatitis virus glycoprotein G (VSV-G) and the baculovirus gp64[28,29]. In the gB trimer, DI contains the fusion loops, DII has a pleckstrin homology domain, DIII forms a central helical bundle, DIV forms the crown, and DV connects the crown to the transmembrane domain. Our recent single particle cryogenic electron microscopy (cryo-EM) studies of full-length VZV gB have demonstrated the importance of gB DIV in fusion initiation[3,20]. In contrast, the rapid conversion of gB to its postfusion form upon deletion of the transmembrane domain or removal from a lipid bilayer has hindered deriving high-resolution structures of prefusion gB[21,27,30,31]. Extracellular vesicles (EVs) were used as a surrogate for structural analyses of virion envelope-associated glycoproteins, overcoming effects on the native topology of the protein after detergent solubilization and reconstitution into an artificial membrane[32]. Cryogenic electron tomography (cryo-ET) and single particle cryo-EM have revealed prefusion structures for herpes simplex virus 1 (HSV-1) and human cytomegalovirus (HCMV) gB anchored in native membranes[31,33–36]. Expression on EVs of a HSV-1 mutant with a proline substitution at histidine 516 in the DIII central helix yielded a condensed form of gB that allowed assignment of DI-DV within a 9 Å resolution cryo-ET structure[31]. In this structure, DI and DII from each of three protomers created a tripod that encapsulated DIV, which was inverted compared to the postfusion structure, with DIII protruding from the top of the prefusion gB trimer while DV was centrally located and connected the ectodomain via transmembrane helices to the cytoplasmic domain. A 3.6 Å resolution single particle cryo-EM structure of HCMV gB extracted from purified virus particles with an architecture similar to HSV-1 gB[H516P] was achieved using a small molecule fusion inhibitor and chemical cross-linking[36]. Although these structural studies provide information about the stationary phases of herpesvirus gB in pre- and postfusion conformations, the transition between these two forms is not understood.

Here, we demonstrate the critical role of the gB DIII central helix using structure-function studies of gB from the model pathogen VZV. Individual proline substitutions of $^{526}$EHV$^{528}$ within VZV gB DIII inhibits gB/gH-gL mediated cell-cell fusion. Cryo-ET reveals that gB[H527P] is present in the condensed form on the membranes of EVs. Importantly, molecular classification of the gB trimers further reveals two classes of conformations, suggesting a stalled process of conformational change instigated by substitution of the conserved histidine at this position. In addition, the prefusion state of VZV gB[H527P] is supported by binding of an anti-gB neutralizing antibody, mAb 93k, in cells transiently expressing this mutant. Critically, proline substitutions of $^{526}$EHV$^{528}$ in the VZV pOka genome were fully compatible with intracellular assembly of virus particles and their release at cell surfaces, as demonstrated by a flow cytometry-based transmission electron microscopy (FC-TEM) assay. However, only the pOka-gB[E526P] mutant yielded infectious VZV although gB-dependent cell-cell spread is markedly impaired by this mutation. As expected, the fusion-arrested gB[H527P] mutation is lethal to the virus. Inactivation of pOka-gB[V528P] is explained by disrupted ER egress, preventing gB maturation and trafficking to the cell surface, associated with a drastically distorted central helix structure predicted by AlphaFold2. Detection of complete viral particles despite gB ER retention suggested that virus assembly can proceed without gB incorporation into the virion envelope. Thus, the structural integrity of the gB DIII central helix is necessary not only for the molecular transition of gB for fusogenic function but also for essential steps in gB biosynthesis required to produce infectious herpesviruses.

## Results

### The gB[H527P] mutation constrains VZV gB in a prefusion conformation

VZV gB residues, $^{526}$EHV$^{528}$, were within the upper region of the DIII central helix of the HSV-1 gB[H516P]-based prefusion VZV gB homology model and the lower region of DIII in our gB postfusion structure[3,20,31] (Fig. 1a–c). Conservation of the histidine at position 527 across the *alphaherpesvirinae* (Supplementary Fig. 1), suggested that a proline substitution might fix VZV gB in a prefusion form as observed with HSV-1 gB[H516P][31]. The valine at position 528 is conserved in the varicelloviruses, the simplexviruses, and several evolutionary divergent alphaherpesviruses, whereas the glutamic acid at 526 is unique to VZV and one non-human primate varicellovirus (Supplementary Fig. 1). Although gB was detectable in cells transiently transfected with vectors expressing WT gB or proline substitutions at $^{526}$EHV$^{528}$, only WT gB and gB[H527P] were incorporated into EVs (Supplementary Fig. 2a–d). A preliminary cryo-ET dataset revealed that WT gB and gB[H527P] formed different conformations in the membranes of purified EVs from transfected BHK-21 cells (Fig. 1d). WT gB was 160 Å in height, and resembled the 2.8 Å postfusion structure of gB purified from VZV infected cells[3]. DI-IV were discernable, but DV was not, consistent with its location in postfusion gB. In contrast, the gB[H527P] trimers formed condensed structures of 100 Å in height and 100 Å in width that resembled those of HSV-1 gB[H516P], indicating that the H527P substitution prevented the transition of gB to a postfusion conformation structure (Fig. 1d). Notably, WT VZV gB was predominantly in a postfusion conformation, while gB[H527P] was exclusively in a prefusion conformation (Fig. 1d).

To achieve a higher resolution structure of gB[H527P] on EVs, a more extensive cryo-ET dataset was generated and coupled with subtomogram averaging (STA) (Supplementary Fig. 2f; Supplementary Table 1). Clusters of gB[H527P] were evident with edge contacts between the gB molecules on the surface of EVs (Fig. 2a). STA of gB[H527P] on EVs produced an initial cryo-EM map of 10 Å that also revealed intermolecular contacts (Fig. 2b). The overall architecture of gB[H527P] suggested a condensed form similar to HSV-1 gB[H516P] where DI and DII formed a tripod around DIV, which was inverted toward the membrane compared to the postfusion form of gB, and DIII underwent a slight bend and twist that exposed this domain at the top of the trimeric molecule (Fig. 2b). However, the decline of the "gold-standard" Fourier Shell Correlation (FSC) curve at low resolution (20 Å), in addition to the local resolution estimation suggested that large scale conformational heterogeneity might be present in the system (Fig. 2c and Supplementary Fig. 2f). Furthermore, the upper central region of the gB trimer, where DIII was predicted to emerge, had a lower local resolution compared to the rest of the protein (Fig. 2c), also indicative of heterogeneity in the reconstruction. Initial classification of the gB[H527P] particles on the membranes of EVs identified three classes that were randomly distributed in the EV membranes (Supplementary Fig. 2f–h). However, class III was discarded due to low resolution in the center region of the gB trimer. The remaining two classes (class I and class II) with distinct architectures were revealed by further subtomogram classification and focused refinement of the central region of the gB trimer (Fig. 2d and Supplementary Fig. 2f). The separation of the data into two classes improved the consistency of the cryo-EM map resolution (9.8–12.4 Å), which is also evident by the improved shape of the FSC curves (Fig. 2d), suggesting the presence of two predominant gB architectures on the surface of gB[H527P] induced EVs. Molecular dynamics flexible fitting (MDFF) of the VZV gB prefusion model predicted from the HSV-1 gB[H516P] structure was used to fit the domains DI-V into the STA maps for each class (Fig. 2e and f; Supplementary Movie 1). The locations of the $^{526}$EHV$^{528}$ residues in the gB DIII central helix in class I and class II of gB[H527P] was modeled and a slight bend of the central helix was seen similar to that in HSV-1 gB[H516P] (Supplementary Fig. 3a).

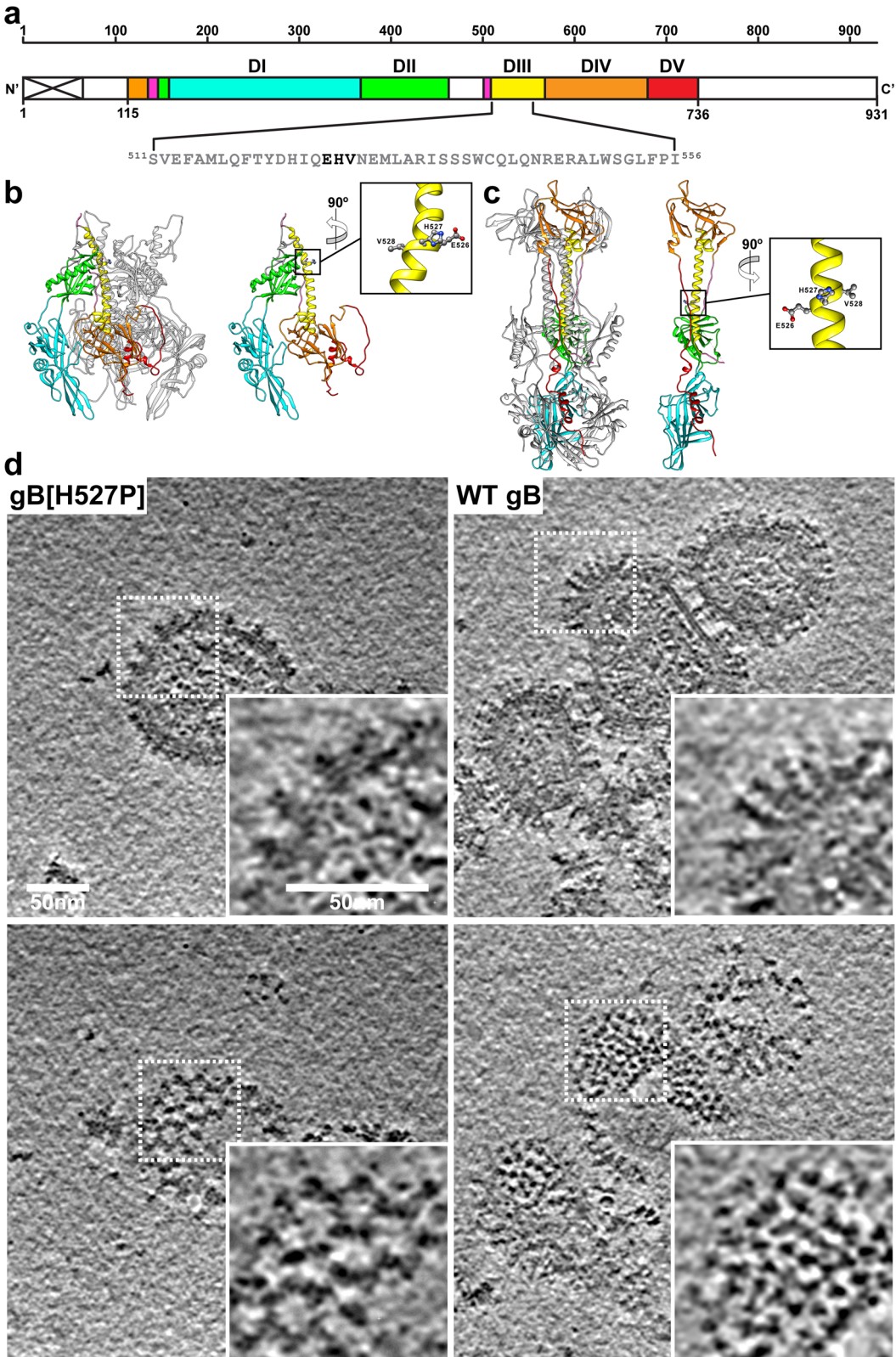

**Fig. 1 | The point mutation H527P constrains VZV gB in a prefusion conformation. a** A diagram of the linear structure for VZV gB highlighting the location of the DIII central helix. The signal sequence is depicted by the crossed white box. Colored regions (residues 115–736) correspond to domains in the cryo-EM structure; DI (cyan), DII (green), DIII (yellow), DIV (orange), DV (red) and linker regions (hot pink). Amino acids [526]EHV[528] (bold) were targeted for mutagenesis. **b** and **c** Trimer and monomer models for VZV gB prefusion (**b**) by SWISS Model[20] and postfusion (**c**) from the 2.8 Å cryo-EM structure[3]. The inset shows the [526]EHV[528] site in the DIII central helix. **d** Single plane views of representative tomographic reconstructions (see Supplementary Table 1) from cryo-ET of EVs produced by BHK-21 cells transfected with gB[H527P] or WT gB expression vectors. Upper and lower panels are different planes from the same tomogram to show side and top views of gB[H527P] or WT gB expressed on EVs. Boxes with dashed lines are magnified areas shown in the insets. Scale bars shown are for each panel and inset.

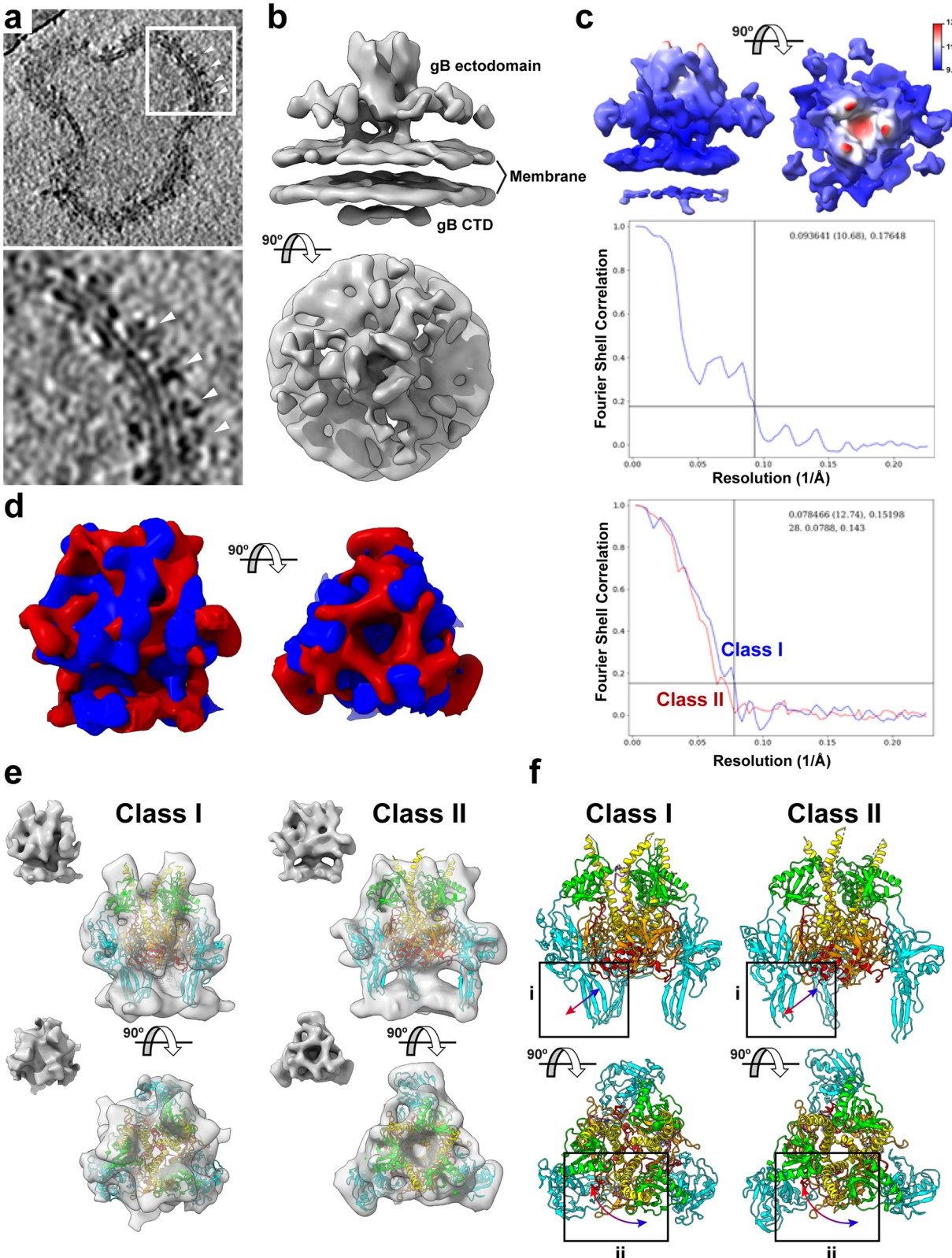

Although the tripod shape of the condensed form of gB was retained, there was considerable lateral movement of the gB DI fusion loop region (Fig. 2e and f box i; Supplementary Movie 1) and rotation of gB DIII between the two classes (Fig. 2e and f box ii; Supplementary Movie 1) as predicted by the MDFF of the two classes. The modeled trajectories for DI and DIII between the two classes paralleled the low resolution observed for the discarded class III, suggesting the presence of further intermediate structures transitioning between the two predominant classes of VZV gB[H527P]. Thus, the substitution of proline at gB H527 prevents the transition of VZV gB to a postfusion conformation but allows movement of the gB trimer between a closed (class I) and open (class II) condensed form. These data provide evidence for the potential transitional steps from a pre- to postfusion conformation.

**Fig. 2 | Cryo-ET structures of VZV gB[H527P] resident on the membranes of EVs. a and b** A representative slice view of a tomogram (**a**; see Supplementary Table 1) and an EMAN2 cryo-EM map generated by subtomogram averaging (STA) (**b**) from EVs purified from BHK-21 cells transiently transfected with gB[H527P]. The white box in the upper panel of (**a**) highlights the area shown in the lower panel. Arrowheads point to gB[H527P] on the membrane of EVs. **c** A 10 Å resolution cryo-EM map derived from a local refinement of a tight mask and the associated Fourier Shell Correlation (FSC) curve. The coloring represents resolution from 9.8 Å (blue) to 12.4 Å (red). **d** STA maps and FSC curves for two classes (I – blue; II – red) of gB[H527P] identified on the surface of EVs using EMAN2. **e** STA maps of Class I and Class II gB[H527P] molecules (inset) with each panel showing MDFF of the homology model of the VZV gB[20] based on HSV-1 gB[H516P][31] into the cryo-EM maps. Snapshots from Supplementary Movie 1 are shown, the VZV gB domains are colored as for Fig. 1; DI (cyan), DII (green), DIII (yellow), DIV (orange), DV (red) and linker regions (hot pink). **f** Molecular models from the MDFF of the VZV gB homology model with boxed areas (i and ii) highlighting regions of structural differences between the two models. Arrows (blue to red gradient) represent the proposed movement between the class I and class II maps.

## The prefusion gB[H527P] maintains the epitope of a neutralizing mAb

Because production of EVs from gB[E526P] was limited and gB[V528P] yielded none, effects of proline substitutions on the DIII central helix structure were modeled using AlphaFold2 (Supplementary Table 2). Both E526P and H527P induced a modest deformation whereas V528P was predicted to bend the helix by almost 90 degrees compared to WT gB DIII (Fig. 3a). Alanine substitutions of [526]EHV[528] modeled to assess the contribution of E, H, and V side chains to DIII structure did not significantly affect the central helix architecture (Fig. 3a).

The functional consequences of changes in DIII architecture due to [526]EHV[528] proline mutations was assessed using the human VZV neutralizing mAb 93k. Our previous 2.8 Å cryo-EM structure of postfusion VZV gB in complex with mAb 93k revealed a conformational epitope mapped to gB DIV (the crown)[3]. Neutralization of VZV infection by mAb 93k via fusion inhibition indicates accessibility of this epitope on prefusion gB, which was supported by our VZV gB prefusion homology model[3,20]. When the accessibility of this functional epitope was quantified by flow cytometry to determine the frequency of transiently transfected cells expressing the gB [526]EHV[528] mutants and median fluorescence intensity (MFI) with levels of gB expression normalized to WT gB (Fig. 3b and Supplementary Fig. 4), the frequency of gB-expressing cells was comparable to WT gB for all proline and alanine mutants with the exception of gB[V528P] (Fig. 3c). The frequency of cells expressing the [526]EHV[528] proline mutants was consistent with the relative total protein abundance detected in western blots (Supplementary Fig. 2b), reflecting the steady state level of those three mutant proteins. The gB[E526P] mutant had reduced MFI compared to WT gB (Fig. 3c), suggesting that this mutation might generate an allosteric conformational change in the epitope even though the DIII central helix structure was predicted to be affected only modestly (Fig. 3a). MFI for gB[V528P] was markedly reduced, attributable to disruption of the mAb 93k epitope due to gB misfolding caused by drastic distortion of the helix (Fig. 3a), the significantly decreased abundance of this mutant protein (Fig. 3c and Supplementary Fig. 2b), or likely both. Notably, 93k mAb binding to gB[H527P] was comparable to WT gB (Fig. 3c), indicating preservation of the conformational 93k epitope and confirming that the gB[H527P] cryo-EM structures represent forms of gB locked in prefusion conformations. Consistently, the 93k epitope was accessible when Fab binding was modeled for the class I and class II gB[H527P] based on 93k footprint on the gB DIV postfusion structure (Supplementary Fig. 3b).

## Targeted mutagenesis of gB [526]EHV[528] disrupts gB fusion function

Given the structural changes in gB imposed by gB[H527P] and predicted for gB[E526P] and gB[V528P] (Figs. 2 and 3a), together with the differences in 93k binding, the effect of these mutations on membrane fusion was assessed using a stable reporter fusion assay (SRFA). This assay measures gB/gH-gL mediated fusion as a surrogate for membrane fusion during virus entry and cell-cell fusion in the absence of other VZV proteins[6,14–17]. The fusion activity for all three proline substitutions was barely detectable compared to WT gB (Fig. 3d; Table 1). Failure to induce gB/gH-gL fusion was consistent with the inability of gB[H527P] to transition to a postfusion

conformation. Alanine mutants, gB[E526A], gB[H527A] and gB[V528A], retained fusion activity of approximately 48%, 90% and 68% respectively compared to WT gB (Fig. 3d; Table 1). These data indicate that the DIII central helix architecture is necessary whereas the amino acid side chain interactions of [526]EHV[528] support but are not critical for gB/gH-gL mediated fusion. This was consistent with minimal deformation of DIII by [526]EHV[528] alanine substitutions predicted by AlphaFold2 (Fig. 3a) and supported by the preservation of mAb 93k epitope (Fig. 3b, c).

## VZV is inactivated by gB[H527P] and gB[V528P] but not gB[E526P]

To determine the role of gB DIII in VZV replication and cell-cell spread, proline or alanine substitutions at [526]EHV[528] were transferred to pOka-TKGFP Bacterial Artificial Chromosomes (BACs). As expected, infectious VZV was recovered from MeWo cells transfected with BACs containing the fusion competent alanine substitutions, gB[E526A], gB[H527A] and gB[V528A], and plaque sizes were comparable to the BAC with WT gB (Fig. 3e, f; Table 1). However, for the proline substitutions, only pOka-TKGFP-gB[E526P] yielded infectious VZV (Fig. 3e; Table 1). Although plaque sizes of pOka-TKGFP-gB[E526P] were significantly smaller than WT pOka-TKGFP plaques (Fig. 3f), expression of this gB mutant allowed the production of virus particles that could enter neighboring cells even though gB fusion function was severely impaired (Fig. 3d; Table 1). Infectious VZV was not recovered from the pOka-TKGFP-gB[H527P] or pOka-TKGFP-gB[V528P] BACs despite five separate transfections and at least six passages to detect residual function or emergence of compensatory mutations.

Like other herpesviruses, VZV genes are expressed at immediate early (IE), early (E), and late (L) times after infection. The essential IE protein, IE62, thymidine kinase, an E protein, and the L proteins ORF23, a minor capsid protein, and ORF31 (gB), were all produced in MeWo cells transfected with the pOka BAC and the mutant BACs with proline substitutions of [526]EHV[528] (Fig. 4a and Supplementary Fig. 5a). To investigate whether VZV particle assembly was disrupted by the inactivating mutations, gB[H527P] and gB[V528P], VZV particle morphogenesis was assessed using a flow cytometry-based transmission electron microscopy (FC-TEM) assay; VZV infected cells were enriched by cell sorting based on the GFP fluorescence signal of TKGFP (Fig. 4b). Nucleocapsids were observed in pOka-TKGFP-gB[E526P] and pOka-TKGFP-gB[H527P] BAC transfected cells at 48 hours with morphologies like those present in pOka-TKGFP BAC transfected cells, including viral DNA containing C-capsids, and scaffold containing B-capsids or empty A-capsids (Fig. 4c), while none were found in pOka-TKGFP-gB[V528P] BAC transfected cells (Fig. 4c). Herpesviruses produce complete particles with viral DNA containing capsids and light particles that are enveloped and contain tegument proteins but no capsids. Both light and complete particles were observed on the surfaces of pOka-TKGFP-gB[E526P], pOka-TKGFP-gB[V528P] and pOka-TKGFP BAC transfected cells (Fig. 4c), demonstrating that proline substitutions at E526 and V528 in gB DIII did not inhibit secondary envelopment of capsids or particle egress across the plasma membrane. The complete virus particle on the surface of the pOka-TKGFP-gB[V528P] BAC transfected cell confirmed that

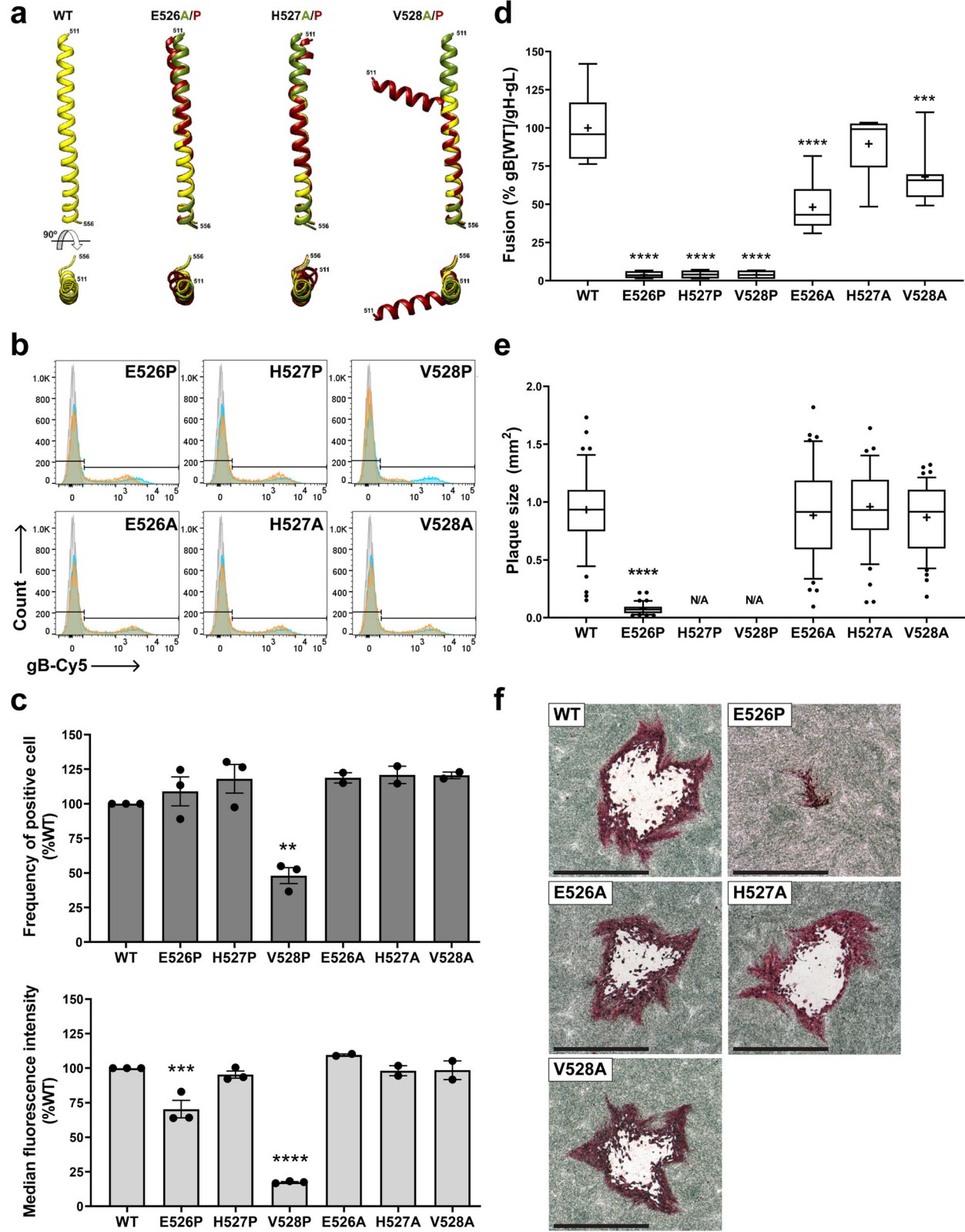

nucleocapsids could be produced despite the mutation. Notably, a genome-containing particle was identified in the cytoplasm of pOka-TKGFP-gB[H527P] BAC transfected cells (Supplementary Fig. 5b), together with the light particles observed (Fig. 4c), indicating that gB[H527P] did not interfere with virus particle assembly. Thus, enveloped VZV particles could be assembled even when gB mutations prevented VZV replication.

### Disrupted biosynthesis renders gB[V528P] fusion defective

To further dissect the role of the gB DIII central helix, the intracellular localization of gB $^{526}$EHV$^{528}$ mutants was evaluated by confocal microscopy after transient expression. Mutants with alanine substitutions were compared to WT gB, which traffics from the endoplasmic reticulum (ER) to the trans-Golgi network (anti-TGN46) and has a punctate distribution within early endosomes (anti-EEA1), as we reported

**Fig. 3 | Targeted mutagenesis of gB [526]EHV[528] disrupts gB/gH-gL dependent membrane fusion and viral replication. a** The predicted structures for the WT and mutant VZV gB DIII central helices by AlphaFold2[63,64] and the mutant structures aligned to WT gB using Matchmaker[62] are shown; WT gB: yellow, alanine mutants: green, proline mutants: red. **b** Flow cytometry analysis of CHO-DSP1 cells transfected with vectors expressing WT gB or gB proline and alanine mutants, permeabilized, and immunostained with Cy5-conjugated mAb 93k; nonspecific antigen control: gray, WT gB: blue, mutant gB: orange. Gating strategy is provided in Supplementary Fig. 4a. **c** Frequency of gB-expressing cells detected by mAb 93k and median florescence intensity (MFI) of mAb 93k binding normalized to WT gB (mean ± SEM). n = 3 independent experiments for WT gB and proline mutants, and n = 2 independent experiments for alanine mutants. Statistical differences were evaluated by one-way ANOVA, Dunnett's multiple comparisons test (**, $p = 0.0014$; ***, $p = 0.0004$; ****, $p < 0.0001$). **d** Cell-cell fusion measured by the stable reporter fusion assay (SRFA) using CHO-DSP1 cells transfected with vectors expressing WT gB or gB mutants together with gH-gL expression vectors, mixed with MeWo-DSP2

reporter cells. Fusion efficiency was normalized to that mediated by gB[WT]/gH-gL (%). Data from n = 8 independent experiments is shown in Box (25-75 percentile) and Whisker (10-90 percentile) plots; horizontal bar is the median and the + indicates the mean. Statistical differences were analyzed by one-way ANOVA, Dunnett's multiple comparisons test (***, $p = 0.0005$; ****, $p < 0.0001$). **e** Plaque sizes in MeWo cells infected with pOka (WT), pOka-TKGFP-gB[E526P], pOka-TKGFP-gB[E526A], pOka-TKGFP-gB[H527A], or pOka-TKGFP-gB[V528A] at 4 dpi; lethal mutant viruses pOka-TKGFP-gB[H527P] and pOka-TKGFP-gB[V528P] are indicated by N/A (not available). Comparison of 45 plaques per virus (mm²) from n = 3 independent experiments is shown by Box (25-75 percentile) and Whisker (10-90 percentile) plots; dots are outliers, horizontal bar is the median and the + indicates the mean. Statistical differences between plaque sizes of WT and mutant viruses were evaluated by one-way ANOVA, Dunnett's multiple comparisons test (****, $p < 0.0001$). **f** Immunohistochemistry staining of VZV plaques in MeWo cells; Representative examples for each virus plaque recorded 45 times from n = 3 independent experiments are shown. Scale bar = 1 mm. Source data are provided as a Source Data file.

previously[4,7] (Fig. 5a). The proline substitutions at E526 and H527 exhibited expected localization compared to WT gB, whereas gB[V528P] failed to move to either the TGN or early endosomes (Fig. 5a), indicating severe disruption of gB intracellular trafficking.

Next, the effects of the [526]EHV[528] mutations on gB synthesis and steps in its post-translational maturation were compared to WT gB using glycosylase endoglycosidase H (Endo H), which removes high-mannose N-linked glycans added to proteins in the ER but not the complex glycan modifications acquired in the TGN; peptide-N-glycosidase F (PNGase F) removes all N-linked glycans. Without treatment, full-length gB was -124 kDa, and the furin-cleaved C-terminal fragments were 66 kDa and 60 kDa (Fig. 5b). Endo H reduced full-length gB to approximately 95 kDa, the same size as gB treated by PNGase F, indicating ER localization of the full-length immature gB. The C-terminal cleavage fragments were Endo H resistant, with molecular weights of 62 kDa and 50 kDa, whereas the fragment treated by PNGase F was less than 50 kDa, indicating that proteolytic cleavage during maturation occurred after ER egress (Fig. 5b). In particular, the detection of a single C-terminal gB fragment after PNGase F treatment implies that the 66 kDa and 60 kDa fragments for untreated gB are differentially N-linked glycoforms and not distinct cleavage products.

The alanine substitutions of [526]EHV[528] and proline substitutions at E526 and H527 did not alter effects of Endo H or PNGase F treatment compared to WT gB, indicating that protein synthesis and post-translational modification were not significantly disrupted (Fig. 5b). In contrast, only immature, ER-associated gB was produced by the gB[V528P] mutant and gB was substantially reduced in quantity compared to WT gB (Fig. 5b and Supplementary Fig. 2b). Absence of the C-terminal cleaved glycoforms showed gB[V528P] was not transported out of the ER for further processing and maturation. These data were consistent with its absence on EVs (Supplementary Fig. 2a, b) and significantly reduced abundance on cell surfaces (Supplementary Fig. 4b, c).

Disruption of the canonical biosynthesis of gB, required for gB incorporation into the envelope of VZV particles, explained why gB[V528P] did not induce cell fusion (Fig. 3d; Table 1) and pOka-TKGFP-gB[V528P] failed to replicate (Fig. 3e; Table 1). Since gB[V528A] did not differ from WT gB, the consequences are not due to an amino acid side chain change but can be attributed to distortion of the gB DIII central helix structure by gB[V528P] as predicted by AlphaFold2 (Fig. 3a). The amino acids equivalent to V528 in VZV gB were relatively conserved among all the human herpesviruses (Supplementary Fig. 6a) and AlphaFold2 predicted a similar distortion in the gB DIII central helix by proline substitutions at this position (Supplementary Fig. 6b; Supplementary Table 3) except for HHV8 gB, which required an additional G471Y mutation to allow the bend caused by I472P (Supplementary Fig. 6b; Supplementary Table 3) and all betaherpesviruses have a tyrosine residue equivalent to G471 (Supplementary Fig. 6a). The predicted deformation by proline substitution to valine or

isoleucine is a conserved feature, supporting the important role of the equivalent V528 position in maintaining the helical structure of herpesvirus gB DIII.

## Discussion

Like other viral fusogens, a conformational change of herpesvirus gB from a pre- to postfusion structure is thought to drive fusion of the virion envelope with the host cell membrane for entry[2]. This is based on end-point structures for the energetically stable postfusion gB and for gB stabilized by either mutagenesis of DIII or using a fusion inhibitor and chemical cross-linking[3,21–27,31,36]. Identifying transitional states of the trimer is essential for understanding the molecular dynamics of gB as a fusogen for rational drug design and vaccine development. The present study visualizes two such presumed transitional states using cryo-ET STA and focused classification of the VZV gB mutant, H527P. The closed (class I) and open (class II) conformations of VZV gB[H527P] and the MDFF fitting of a prefusion homology model based on HSV-1 gB[H516P] into the two cryo-EM STA maps provide insights into the first steps in this dynamic process.

A critical observation about the related HSV-1 gB[H516P] mutant was that the soluble ectodomain transitions to a postfusion conformation as shown by X-ray crystallography, demonstrating that this mutation does not fully lock gB[31]. This supports the concept that prefusion gB stability requires the transmembrane domain and C-terminal domain (CTD) along with the lipid membrane[21,27,30,31]. Consistent with this notion, the CTD of VZV gB contains elements that regulate membrane fusion[4,7,37]. In addition, herpesvirus gB requires the gH-gL heterodimer for its fusion function and regulation[2,38,39]. Whether gH-gL relays a gB activation signal from cellular receptors or other viral proteins, to trigger gB-mediated membrane fusion, and whether gH-gL acts as a "gatekeeper" to spatially and temporally regulate gB from premature fusion remain active areas of investigation. Our previous studies have demonstrated the likely involvement of the gH-gL heterodimer to regulate VZV gB fusion; truncation of the VZV gH CTD yields a hyperfusogenic phenotype detrimental for skin replication[6]. Similar findings were reported for HSV-1 where truncation of the gH CTD also increased fusion[10,40]. The regulation of gB by gH-gL is further supported by the physical interaction of the trimer with the heterodimer[35,41]. Critically, for VZV, the present study demonstrated that predominant WT gB on EVs adopts the postfusion conformation in the absence of gH-gL, providing further evidence that gH-gL is required to maintain VZV gB in a prefusion conformation. Although 70% of WT HSV-1 gB was reported to remain in a prefusion conformation on EVs in the absence of gH-gL, the gB[H516P] mutation was still required to stably lock the trimer in a prefusion conformation on EVs[31]. Together, these data support our observations that without the constraints of gH-gL, the VZV gB[H527P] is caught between to kinetically trapped conformations, presumably early transitions states (Supplementary Fig. 7).

**Table 1 | Properties of the mutant protein and recombinant viruses bearing the proline and alanine substitutions to $^{526}$EHV$^{528}$ in VZV gB DIII central helix**

| Mutant | Protein | | Recombinant virus | | | | |
|---|---|---|---|---|---|---|---|
| | Fusion activity[a] | Intracellular localization[b] | Protease cleavage[b] | EVs production | Viral gene expression | TEM morphology | Growth phenotype |
| E526P | 3.8% | ER, trans-Golgi, early endosome | Yes | No | IE, E, L | Capsids; virus particles | Severely impaired; small plaque |
| H527P | 4.1% | ER, trans-Golgi, early endosome | Yes | Yes | IE, E, L | Capsids; virus particles | Lethal |
| V528P | 3.8% | ER | No | No | IE, E, L | virus particles | Lethal |
| E526A | 48.0% | ER, trans-Golgi, early endosome | Yes | ND | ND | ND | WT |
| H527A | 89.6% | ER, trans-Golgi, early endosome | Yes | ND | ND | ND | WT |
| V528A | 67.9% | ER, trans-Golgi, early endosome | Yes | ND | ND | ND | WT |

[a]Fusion activity was measured by SRFA and normalized to gB[WT]/gH-gL (%).
[b]WT gB localizes to ER, trans-Golgi network, and early endosomes and undergoes cleavage.
EVs extracellular vesicles, IE Immediately Early gene, E Early gene, L late gene, ND experiment was not performed.

The two VZV gB[H527P] classes imply that the previously reported prefusion structures for HSV-1 gB[H516P] or HCMV might not represent the complete array of prefusion gB conformations. The architecture of the HSV-1 gB[H516P] structure is similar to the more open conformation of the class II structure for VZV gB[H527P] in the present study[31]. In addition, the soluble, truncated HSV-1 gB[H516P] ectodomain forms a postfusion structure, inferring that the full-length proline mutant only becomes trapped during its transition in the context of the membrane[31]. Furthermore, HCMV gB was captured using a thiourea fusion inhibitor, WAY-174865, and chemical cross-linking. WAY-174865 binds to the gB fusion loops, membrane proximal region (MPR), and transmembrane domain (TM) through interactions with hydrophobic and aromatic residues, including the critical fusion loop residues Y153 and Y155[36]. Importantly, the WAY-174865 binding pockets in the HCMV gB trimer are located internally in between the α8 and α9 glycine-rich amphipathic helices, indicating that the gB trimer must be in an open configuration for the compound to gain access to the MPR, TM, and fusion loops as seen for the 3.6 Å resolution cryo-EM structure[36]. While the activity data for WAY-174865 is not available, previous studies on two closely related thiourea compounds, CF102 and WY1768, demonstrated that these HCMV fusion inhibitors do not stably interact with cell-free virus particles and are only effective once virus particles bind to the cell, preventing a complete conformational change of gB triggered by receptor interaction[42,43]. CF102 inhibition escape mutations have also been identified in the WAY-174865 binding pocket, including α9 amphipathic helix[36], providing structural evidence for the shared mechanism of action of WAY-174865 and related thiourea compounds. These studies indicate that thiourea fusion inhibitors target exposed regions in gB transitional states rather than a more closed prefusion conformation similar to VZV gB[H527P] class I in the present study. This concept was proposed for HCMV gB where a "snapshot" of a "breathing" molecule was locked into an intermediate conformation by the fusion inhibitor and by the cross-linking agent[44]. Together, this supports the notion that additional gB prefusion conformations can populate herpesvirus envelopes.

The absence of gB[E526P] on EVs and the predominance of immature gB in transfected cells (Supplementary Fig. 2a, b) was consistent with a moderate decrease in cell surface trafficking (Supplementary Fig. 4b,c), which might contribute to disrupted gB/gH-gL mediated cell-cell fusion activity. In addition, although the effect of E526P on the central helix structure was minimal as predicted by AlphaFold2, the mutation might induce an allosteric change in DIV since binding of neutralizing antibody mAb 93k to its conformational DIV epitope was reduced (Fig. 3c). The barrier to gB/gH-gL dependent fusion is higher under cell-cell fusion assay conditions compared to VZV infected cells in culture, as observed with gB[E526P] inhibition of cell-cell fusion but recovery of pOka-TKGFP-gB[E526P] virus and plaque formation, albeit limited. These findings corroborated the importance of assessing the contributions of gB/gH-gL to fusion function by testing mutations in the viral genome. Furthermore, although pOka-TKGFP-gB[E526P] produced infectious virus, the marked reduction in plaque sizes was in the range associated with failure of VZV to replicate in human skin xenografts in the SCID mouse model of VZV pathogenesis, supporting the conclusion that even slight deviations in the gB central helix will be detrimental for herpesviruses as pathogens in their native hosts[4,6,14].

In line with the severe DIII central helix distortion predicted by AlphaFold2, the proline substitution at V528 had the greatest affect as gB[V528P] did not exit the ER, severely limiting intracellular trafficking necessary for gB maturation required to execute its fusion function. However, while lethal for replication, defective biogenesis of gB[V528P] was compatible with VZV particle assembly, showing that morphogenesis can occur independently of gB maturation and trafficking, as reported for other herpesviruses[45,46]. The fate of gB[V528P] was likely proteolysis as proteins misfolded

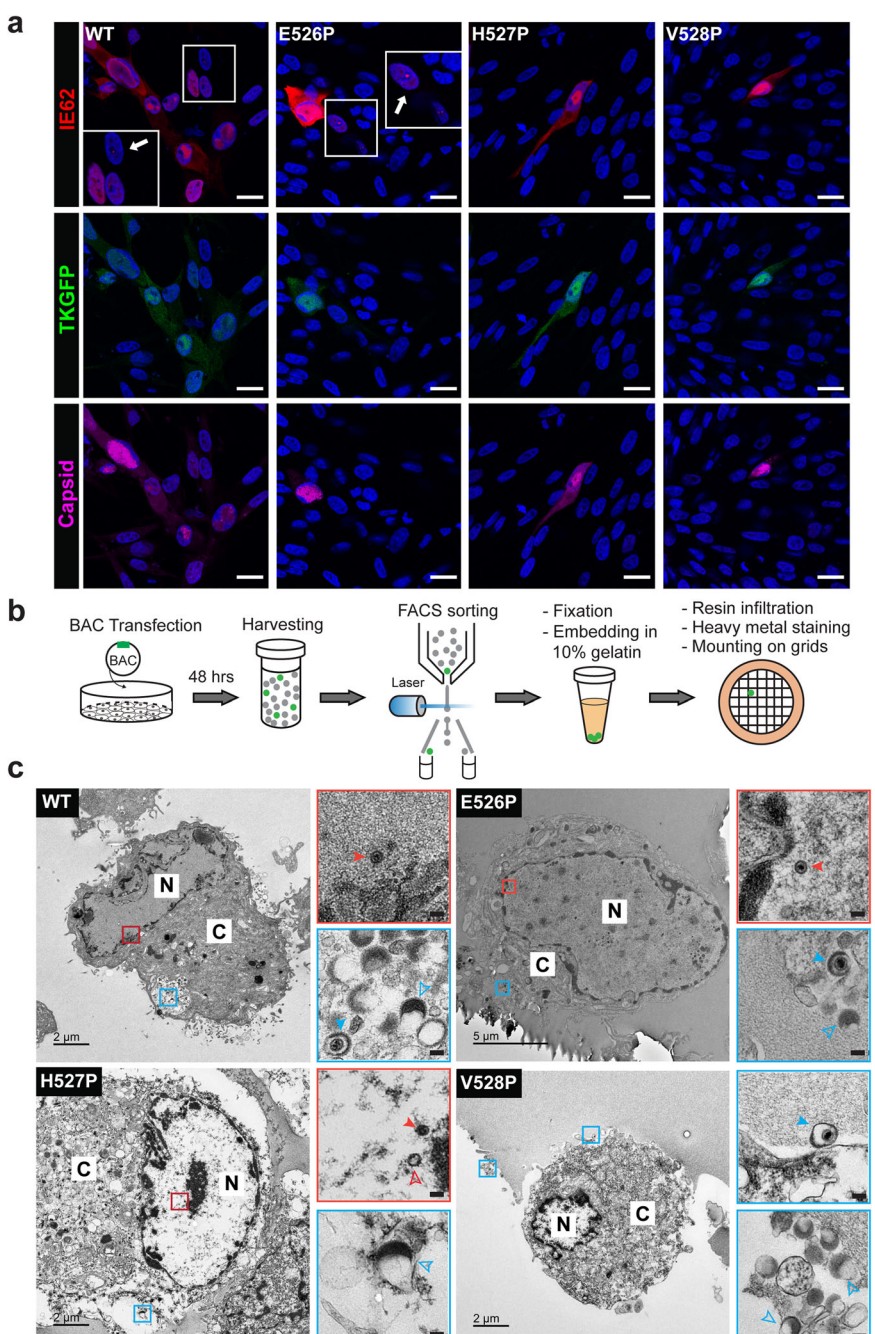

**Fig. 4 | Disrupted gB fusion function did not interfere with VZV immediate early, early or late protein synthesis or with virus particle assembly. a** MeWo cells transfected with pOka-TKGFP (WT), pOka-TKGFP-gB[E526P], pOka-TKGFP-gB[H527P], or pOka-TKGFP-gB[V528P] BACs were imaged by confocal microscopy at 48 hrs post transfection. Immediate-early protein (IE62, red), early protein, thymidine kinase (TK) expressed as a GFP fusion protein (green) and late protein (capsid-ORF23, violet); nuclei stained with Hoechst 33342 (blue). Areas highlighted by a small white box are magnified in the larger white boxes, white arrows indicate cells at the early stage of infection identified by punctate expression of IE62 in nuclei with little to no TKGFP expression. Representative examples from 5 fields of view recorded from n = 2 independent experiments are shown. Scale bar = 20 μm. **b** Diagram of the flow cytometry-based transmission electron microscopy (FC-

TEM) assay. MeWo cells were transfected with VZV BACs expressing TKGFP, harvested at 48 hrs post transfection, and sorted for GFP-positive cells; cells were fixed, embedded in 10% gelatin, infiltrated with resin, heavy metal staining and mounted on the grid. **c** TEM images of MeWo cells transfected with pOka-TKGFP (WT), pOka-TKGFP-gB[E526P], pOka-TKGFP-gB[H527P], or pOka-TKGFP-gB[V528P] BACs. Panels at the right are magnifications of areas indicated by boxes (N: nuclei; C: cytoplasm). Boxes outlined in red show representative viral capsids, with solid red arrowheads indicating viral DNA containing C-capsid and open red arrowheads indicating empty A-, or scaffold containing B-capsids. Boxes outlined in blue show representative viral particles, with solid blue arrowheads indicating complete virus particles and open blue arrowheads indicating light particles. Assay efficiency is provided in Supplementary Table 4. In the magnified images, scale bar = 0.1 μm.

in the ER eventually translocate to the proteasome for degradation[47]. V528 is fully conserved in all alphaherpesvirus orthologues of gB except for MeHV-1 (Meleagrid alphaherpesvirus 1), GaHV-2/3 (Gallid alphaherpesvirus 2/3), and ChHV-5 (Chelonid alphaherpesvirus 5), which have a conservative replacement to

isoleucine, and the distantly related TeHV-3 (Testudinid alphaherpesvirus 3) which has an asparagine (Supplementary Fig. 1). The betaherpesviruses and gammaherpesviruses have a conserved change to isoleucine at the residue equivalent to V528 (Supplementary Fig. 6a). A similar severe distortion in DIII was predicted

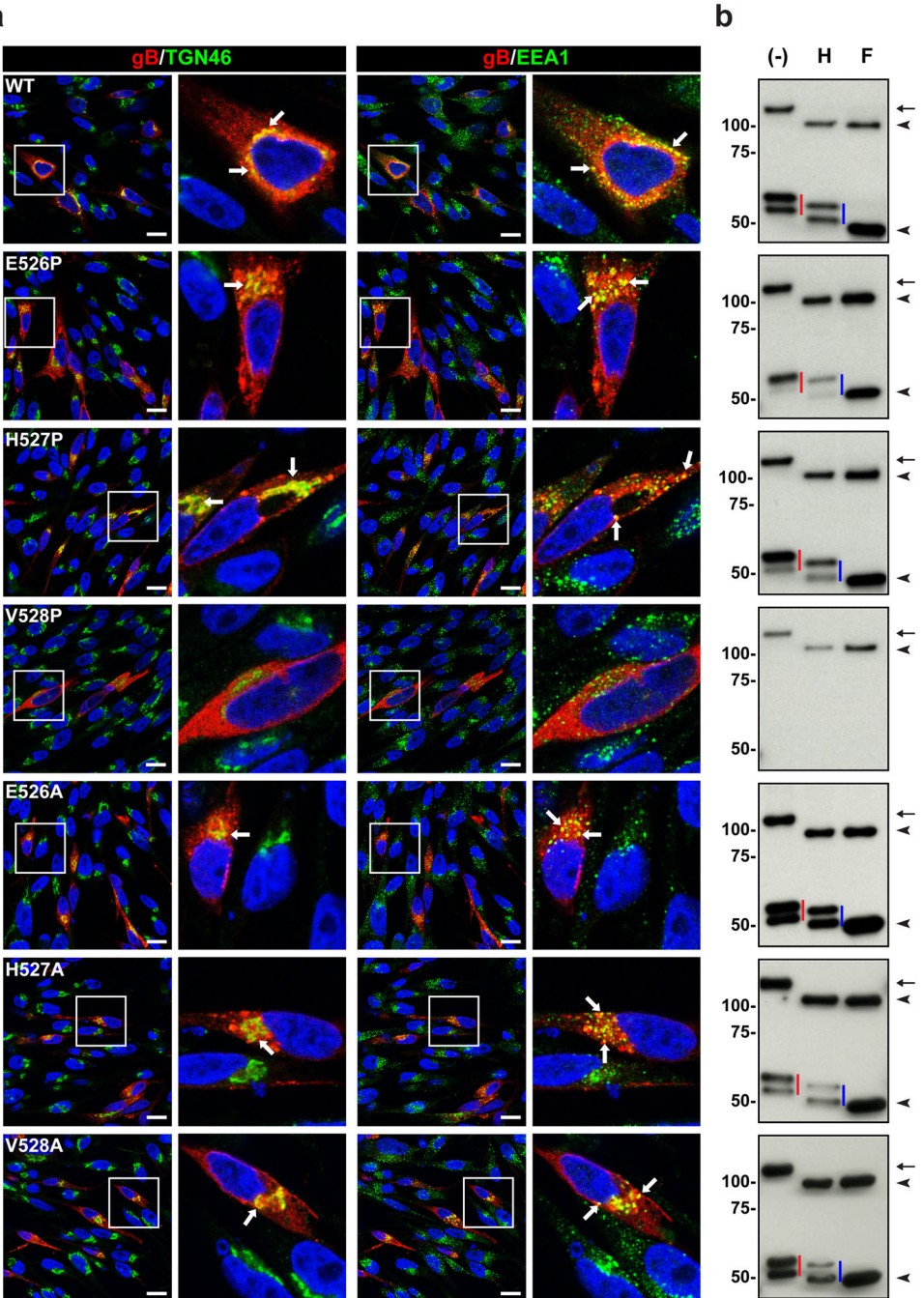

**Fig. 5 | Canonical biosynthesis of gB was disrupted by V528P. a** MeWo cells transiently expressing WT gB and gB mutants were imaged by confocal microscopy at 24 hrs post transfection. Cells were stained with mouse anti-gB mAb SG2 (red), trans-Golgi network marker, TGN46 (green), or early endosome antigen, EEA1 (green), and nuclei were stained with Hoechst 33342 (blue). Representative images are from experiments performed at least in duplicate. Areas in white boxes are shown at higher magnification in panels to the right; white arrows indicate gB colocalization with TGN46 or EEA1 respectively. Representative examples from 5 fields of view recorded from n = 3 independent experiments for proline mutants and n = 2 independent experiments for alanine mutants are shown. Scale bar = 15 μm. **b** Intracellular trafficking of gB was assessed by glycosylation status. MeWo cells transfected with vectors expressing WT gB, or gB mutants were lysed at 24 hrs post transfection and immunoprecipitated with anti-gB mAb SG2, left untreated (-), or treated with Endo H (H) or PNGase F (F). Proteins were separated by SDS-PAGE gel (7.5%) under reducing conditions, and gB was detected with rabbit anti-gB 746-867. The arrows and red vertical lines indicate the glycosylated forms of full-length and cleaved gB respectively, the blue vertical lines indicate the Endo H-resistant glycoforms of cleaved gB, and arrowheads indicate forms of gB deglycosylated by both Endo H and PNGase F. Representative examples from n = 2 independent experiments are shown. Mass markers (kDa) are shown on the left. Source data are provided as a Source Data file.

for the equivalent gB proline substitution in all human herpesvirus, implying a critical role in gB biogenesis likely conserved for the *Herpesviridae.*

The induction of neutralizing antibodies that inhibit virus entry is a cornerstone of vaccine design to protect against disease caused by enveloped viruses, first achieved with live attenuated vaccines. Since neutralizing antibodies that target epitopes accessible on the prefusion conformation of viral fusogens might be more potent, prefusion immunogens have become a strategy for subunit vaccine design. Proline substitutions targeting the helical structure of the

protein (i.e., the heptad repeat in the coiled-coil region) have been used to stabilize several class I fusion proteins for evaluation as candidate vaccines, including the HIV-1 envelope, the respiratory syncytial virus fusion protein, and the spike proteins of the coronaviruses, Middle East respiratory syndrome (MERS)-CoV, severe acute respiratory syndrome coronavirus (SARS-CoV), and SARS-CoV-2[48–52]. Our work indicates that proline stabilized VZV gB has the potential to elicit neutralizing antibodies predominantly to prefusion gB and impair the transition to the postfusion conformation effectively rendering VZV incapable of membrane fusion and cell entry. Due to the highly conserved nature of gB, such an approach in vaccine design could be applicable to all herpesviruses to reduce the burden of disease in animal or human populations.

## Methods

### Cells

CHO-DSP1 cells[17], derived from Chinese hamster ovary (CHO) K1 cells (CCL-61, ATCC) that express Dual Split Protein (DSP)$_{1-7}$ of the chimeric reporter protein composed of split GFP and split *Renilla* luciferase, were propagated using F-12K nutrient mixture medium (Corning) supplemented with 10% fetal bovine serum (FBS, Gibco), penicillin (100 U/ml, Gibco), and streptomycin (100 μg/ml, Gibco). Human melanoma MeWo cells (HTB-65, ATCC) and MeWo-DSP2 derived from MeWo cells that express DSP$_{8-11}$[17] were propagated in minimal essential medium (MEM, Corning) supplemented with 10% FBS, nonessential amino acids (1×, Corning), penicillin (100 U/ml, Gibco), streptomycin (100 μg/ml, Gibco) and amphotericin B (0.5 μg/ml, Corning). CHO-DSP1 and MeWo-DSP2 cells were kept under puromycin selection (8 μg/ml for CHO-DSP1 and 5 μg/ml for MeWo-DSP2; Gibco). Baby hamster kidney (BHK-21) cells were grown in Glasgow minimum essential medium (GMEM) supplemented with 20 mM Hepes pH 7.4, 2% (v/v) TPB and 2% (v/v) FBS.

### Sequence alignment

The amino acid sequence of alphaherpesvirus gB orthologues and human herpesvirus gB orthologues were obtained from the NCBI (https://www.ncbi.nlm.nih.gov/protein). The amino acids that aligned with VZV gB DIII central helix were defined using Clustal Omega (https://www.ebi.ac.uk/Tools/msa/clustalo).

### Construction of gB-expressing vectors with mutations

gB mutant constructs were generated from the pCAGGS-gB[WT] vector, which was a gift from Tadahiro Suenaga and Hisashi Arase (Osaka University, Osaka, Japan)[16]. Primers pairs XmaI-Forward (5′-ATGTG-TAAAGGAGGAAGCCCGGGC-3′)/gB-E526P-Reverse (5′pho-AACATGCGGTTGAATGTGGTCATATG-3′), XmaI-Forward/gB-H527P-Reverse (5′ pho- AACTGGCTCTTGAATGTGGTCATATG-3′), XmaI-Forward/gB-V528P-Reverse (5′ pho-AGGATGCTCTTGAATGTGGTCA-TATG-3′), XmaI-Forward/gB-E526A-Reverse (5′pho-AACATGGGCTTGAATGTGGTCATATG-3′), XmaI-Forward/gB-H527A-Reverse (5′ pho- AACCGCCTCTTGAATGTGGTCATATG-3′), XmaI-Forward/gB-V528A-Reverse (5′ pho-TGCATGCTCTTGAATGTGGTCA-TATG-3′), gB-EHV-Forward (5′pho-AATGAAATGTTGGCACGTATCTCC-3′) /AgeI-Reverse (5′-AAGTTTAAGCACGTACCGGTACGC-3′) containing the desired mutation were used to amplify two PCR products using AccuPrime Taq (Invitrogen). The amplicons were digested with either XmaI or AgeI restriction enzymes (New England Biolabs), and ligation was performed with the digested sticky end and the 5′ phosphate blunt end. Ligated products were electroporated into TOP10F′ Electrocomp E. Coli (Invitrogen) and plated under ampicillin selection. Clones were confirmed by AgeI/XmaI digestion and sequencing using primers gB-pCAGGS-Seq-F (5′-TTCGCGATGAGTATGCACAC-3′) and gB-pCAGGS-Seq-R (5′-TTGGGTTTCTCGGCAAAGGGATCC-3′).

### Generation of pOka-DX BACs with mutations in gB (ORF31)

The pOka-TKGFP virus was engineered from the parental Oka strain of VZV (pOka) derived from a self-excisable bacterial artificial chromosome (BAC)[53] to express EGFP conjugated to the thymidine kinase (TK)[6]. A pPOKA-ΔORF31-TKGFP BAC lacking ORF31 was constructed previously[4]. Briefly, the shuttle vector, gB-Kan containing ORF31 was digested by BglII and NdeI enzymes (New England Biolabs). pCAGGS-gB plasmids containing the desired mutation were also digested with BglII and NdeI enzymes, and the fragment of interest was inserted into the shuttle vector gB-Kan by ligation, and were electroporated into TOP10F′ Electrocomp E. Coli (Invitrogen) and plated under kanamycin selection. Clones were confirmed by AgeI/XmaI digestion and sequencing using gB-Kan-Seq-F (5′-TAATCATTCCCA-CATCATGGACTGC-3′) and gB-Kan-Seq-R (5′-CCCCTTGCATTACTGTT-TATGTAAG-3′). Then the gB-Kan vector carrying the mutations were used to generate BACs using GS1783 cells containing the pPOKA-ΔORF31-TKGFP BAC by Red recombination[11,53,54]. Fragment insertion was confirmed by PCR using primers [31]F56625-56645 (5′-AGGTA-TAGGCAGTTCCCACGG-3′) and [31]R59697-59717 (5′-TTTCATTGA-GACTTGAAGCGC-3′). All BACs were purified using large-construct purification kit (Qiagen), and digested by HindIII to verify that spurious recombination had not occurred. To generate BAC-derived viruses, MeWo cells were transfected in at least triplicate with two BAC clones for each mutation. Viruses were typically recovered at 5–10 days post transfection and passed on MeWo cells to propagate and titrated on MeWo cells.

### EVs production

BHK-21 cells, at around 70% confluency, were transiently transfected with pCAGGS vectors expressing proline substitutions to $^{526}$EHV$^{528}$, or C-terminal His-tagged WT gB or gB[H527P]. Cells were grown for an additional 48 hours with a media exchange to serum-free GMEM after 24 hours. Vesicles were harvested from the supernatant and cleared from cell debris by centrifugation at 3000 × *g* for 20 min at 4 °C, followed by centrifugation through a 20% sucrose cushion at 100,000 × *g* and resuspended in 20 mM Hepes pH 8, 130 mM NaCl.

### Cryo-ET data collection using the Titan Krios

For grid preparation, 3.5 μL vesicles were mixed with 1.5 μL 5 nm gold fiducials on Quantifoil R2/1 grids and plunge frozen in a propane/ethane mixture after blotting using a manual plunge freezer. Microscopy of WT gB, gB[H527P], and gB[H527P]-His EVs was performed at 300 keV on a Titan Krios electron microscope (Thermo Fisher Scientific) equipped with a Gatan K3 (5 μm/pixel) camera and Bioquantum post-column energy filter controlled using SerialEM[55] for micrograph and automated batch tomography data collection. Movie data (11,283 total stacks) were captured in counted mode with a dose rate of -10.75–11.9 e$^-$/Å$^2$/s and 50 millisecond exposure time per frame and 0.2–0.3 second total exposure time per tilt at a nominal magnification of 81,000 × and a pixel size of 1.095 Å/pixel on the specimen. The defocus was set to −2.0 μm.

### Reconstruction of the cryo-ET maps for gB[H527P]

The motion correction, damage compensation, and tilt stacks for all movie-mode data were generated using WARP[56]. All data processing described was performed using the tomography workflow in EMAN2.99[57,58]. Forty tilt series were automatically reconstructed, and 2,074 particles were manually selected from the tomograms. The CTF was estimated from the tilt images before unbinned, phase-flipped subtomograms were extracted. The initial orientation of each particle was assigned using a vector pointing from the center of the vesicle that the particle was located on, to the coordinates of that particle. The initial model was directly generated from the particles at 30 Å resolution and used as a reference "gold-standard"

subtomogram to perform subtilt refinement on all particles. A soft cylinder mask that includes the protein and membrane density was used during the refinement, and C3 symmetry was applied to the averaged structure. After each iteration of the refinement, local resolution was measured based on the two half-maps, and the averaged structures were filtered according to the local resolution. Then, to resolve the structural heterogeneity at the core of the protein revealed by the local resolution, focused classification was performed based on the orientation determined from the single model refinement, using a mask that covers the central part of the protein. Finally, separate subtomogram and subtilt refinements were performed for the particles of each class from the refinement, resulting in the final averaged structures. The HSV-1-based VZV gB prefusion homology model[20] was fitted into the class I and class II cryo-ET maps by domain restricted Molecular Dynamics Flexible Fitting (MDFF) using Visual Molecular Dynamics (VMD 1.9.4a29; http://www.ks.uiuc.edu/Research/vmd/)[59] to prepare the input and configuration files and the Nanoscale Molecular Dynamics (NAMD 2.14; http://www.ks.uiuc.edu/Research/namd/)[60] software to perform simulations with the CHARMM36 force field[61]. All images and movies were generated using UCSF ChimeraX 1.14[62].

### AlphaFold2 predictions of herpesvirus gB DIII central helix

The gB DIII central helix for eight human herpesvirus gB orthologs was defined by amino acid alignment of the complete gB proteins using Clustal X and comparison to the native VZV gB structure (PDB 6VN1) DIII central helix. AlphaFold2[63,64] was compiled following the installation instructions (https://github.com/deepmind/alphafold) on a Colfax ProEdge DX935 Workstation with an Intel Xeon W-2195 18 C/36 T 2.3Ghz CPU, 4 PNY GeForce RTX2080Ti 11GB GPUs, 256MB RAM, and Centos 8 operating system.

### Stable reporter fusion assays

CHO-DSP1 were transfected with vectors pCAGGS-gB[WT], pCAGGS-gB[E526P], pCAGGS-gB[H527P], pCAGGS-gB[V528P], pCAGGS-gB[E526A], pCAGGS-gB[H527A], or pCAGGS-gB[V528A]; pCAGGS-gB[WT]-His-tagged, or pCAGGS-gB[H527P]-His-tagged, with or without vectors expressing gH-gL, pME18S-gH[TL] and pcDNA3.1-gL using Lipofectamine 2000. At 6 hrs post transfection, the transfected cells were harvested and mixed with MeWo-DSP2 cells for additional 48 hrs. The activity of *Renilla* luciferase was read immediately after adding substrate h-Coelenterazine (Nanolight Technology) on a Synergy H1 Hybrid Multi-Mode Reader (BioTek). Experiments were performed in triplicate at least. The fusion activity for His-tagged WT gB and gB[H527P] was shown in Supplementary Fig. 2e.

### Quantitation of cell surface and total expression of gB

CHO-DSP1 cells were transfected with vectors pCAGGS-gB[WT], or pCAGGS-gB[E526P], pCAGGS-gB[H527P], pCAGGS-gB[V528P], pCAGGS-gB[E526A], pCAGGS-gB[H527A], and pCAGGS-gB[V528A], or vectors expressing gH-gL, pME18S-gH[TL] and pcDNA3.1-gL. At 24 hrs later, cells were dislodged using an enzyme-free cell dissociation buffer (Gibco), fixed with 1% paraformaldehyde (PFA, Boston Bioproducts), and resuspended in FACS staining buffer [PBS with 0.2% IgG-free bovine serum albumin (Jackson ImmunoResearch), and 0.1% NaN₃ (Sigma Aldrich)]. Cell surface gB were detected with primary antibodies mouse anti-VZV gB mAb (93k; human)[3] conjugated with Cy5, at 1:100 dilution. Total amounts of viral glycoprotein production were performed by the same staining protocol, except cells were permeabilized using Cytofix/Cytoperm kit (BD Biosciences) before adding the antibodies and during the following procedure. Stained cells were analyzed using a DXP multi-color FACScan analyzer (Cytek Biosciences), and data were processed with FlowJo (TreeStar) to determine the quantity of total and surface levels of gB respectively. The ratio of cell surface to total quantity was also calculated. Experiments were performed in triplicate for proline substitution mutants and in duplicate for alanine substitution mutants.

### Immunoprecipitation and Western blotting

Whole cell lysate from MeWo cells transiently expressing WT gB or gB with proline and alanine substitutions, or gH-gL, a non-antigen negative control was extracted at 24 hrs post transfection using extraction buffer [0.1 M NaCl, 5 mM KCl (Fisher Scientific), 1 mM CaCl₂ (Fisher Scientific), 0.5 mM MgCl₂ (Fisher Scientific), 1% IGEPAL CA-630 (Sigma Aldrich), 1% Deoxycholate (Sigma Aldrich) in 0.1 M Tris buffer, pH 7.2], with cOmplete EDTA-free protease inhibitor, for 30 min on ice. The sample was centrifuged at $3000 \times g$ for 10 min at 4 °C to get rid of cell debris. The supernatant was immunoprecipitated (IP) with mouse anti-VZV gB mAb (SG2-2E6, 1.15 mg/ml, Meridian Life Sciences) that was crosslinked to protein A beads (Pierce) at 20 µg antibody/30 µl beads slurry. Samples were then either left untreated or treated with Endo H (New England Biolabs) or PNGase F (New England Biolabs) according to the manufacture's protocol. Proteins were denatured under reducing condition and separated by 7.5% SDS-PAGE gel (BioRad), transferred to Immobilon-P PVDF membranes (Millipore) using Trans-blot SD Semi-dry Transfer cell (BioRad) in transfer buffer [48 mM Tris Base (Fisher Scientific), 39 mM Glycine (Fisher Scientific) and 0.01% SDS (Sigma Aldrich), with 20% methanol (Sigma Aldrich)]. The transferred proteins were probed for gB using rabbit polyclonal antiserum, 746-868, that recognizes the epitope [833]PEGMDPFAEKPNAT[846] in the cytoplasmic domain of gB (GenScript Corp, 1:4000)[14], followed by anti-rabbit IgG horseradish peroxidase-linked secondary antibodies (GE Healthcare Life Sciences, UK Ltd., 1:10,000) and Pierce ECL Plus Western Blotting Substrate (ThermoFisher). Chemiluminescence was detected using BioMax MR Film (Carestream). For EVs analysis, after supernatants were removed for vesicle preparations, cells were washed with cold PBS and detached using cell scrapers. Cells were pelleted by centrifugation (5 minutes, $4500 \times g$, 4 °C), transferred into 1.5 ml microfuge tubes and washed in cold PBS again before resuspension in RIPA buffer, 100 µL per T175 flask (50 mM Tris pH 8, 1% NP-40, 0.1% SDS, 150 mM NaCl, 0.5% Sodium Deoxycholate, 5 mM EDTA, 1 mM PMSF). Samples were shaken at 4 °C for 30 min before centrifugation at $500 \times g$ for 10 minutes. Supernatants were mixed in SDS sample buffer and run in parallel with vesicle samples in SDS-PAGE, followed by Coomassie staining or western blotting with the rabbit anti-VZV gB antibody 746-868 (1:4000) followed by anti-rabbit-HRP (Sigma Aldrich Chemie GmbH, 1:10,000). For loading control, western blots were re-probed using a mouse anti-GAPDH antibody (CL3266, Sigma Aldrich Chemie GmbH, 1 µg/ml) followed by anti-mouse-HRP (Sigma Aldrich Chemie GmbH, 1:10,000). The Coomassie staining and Western blotting for His-tagged WT gB and gB[H527P] using rabbit anti-His6 antibody (ab9108, Abcam, 1 µg/ml) was shown in Supplementary Fig. 2c, d. Band density was analyzed using ImageJ.

### Confocal microscopy

MeWo cells transfected with gB-expressing vectors were fixed 24 hrs post transfection with 4% formaldehyde for 10 min. Mouse anti-VZV gB mAb (SG2-2E6, 1.15 mg/ml, Meridian Life Sciences) was used to stain for VZV gB (1:200), and other primary antibodies to cellular proteins were used as markers for early endosomes (EEA1; rabbit polyclonal antibody; NBP1-30914, Novus Biological; 1:200), and the trans-Golgi (TGN46; sheep polyclonal antibody; AHP500G, BioRad; 1:250). Primary antibodies were detected with secondary antibodies, anti-mouse Alexa Fluor 555 (Invitrogen), anti-rabbit Alexa 488 (Invitrogen), and anti-sheep Alexa Fluor 647 (Invitrogen). Nuclei were stained with Hoechst 33342 (Invitrogen). MeWo cells transfected with pOka, pOka-TKGFP-gB[E526P], pOka-TKGFP-gB[H527P], or pOka-TKGFP-gB[V528P] BACs were fixed at 48 hrs or 72 hrs post transfection with 4% formaldehyde for 10 min, and cells were probed using primary antibodies to VZV proteins IE62 (mouse mAb; EMD Millipore MAB8616; 1:200), capsid-

ORF23 (polyclonal rabbit; 1:200)[65] and gB (93k; human mAb; 1:200)[3] and nuclei (Hoechst 33342). Primary antibodies were detected with secondary antibodies, anti-mouse Alexa Fluor 555 (Invitrogen), and anti-rabbit or anti-human Alexa Fluor 647 (Invitrogen). Nuclei were stained with Hoechst 33342 (Invitrogen). Images were captured with a Leica SP8, Whitelight Laser Confocal Microscope. Channel merging and image processing was performed with ImageJ and Photoshop.

### Plaque size assay

Monolayers of MeWo cells ($1 \times 10^5$ cell/cm$^2$) were inoculated with 50 plaque-forming unit (PFU) of cell-associated pOka-TKGFP viruses or pOka-TKGFP-gB[E526P] for 2 hrs, followed by change of medium. Cells were fixed with 4% PFA at 4 days post-infection and processed with Immunohistochemistry (IHC) using mouse monoclonal anti-VZV antibody (Meridian Life Sciences C05108MA, 1:2000), followed by incubation with biotinylated anti-mouse IgG (Vector Laboratories #BA-9200, 1:1000), and alkaline phosphatase streptavidin (Jackson ImmunoResearch #016-050-084, 1:400). The plaques were visualized by staining with a mixture of FastRed salt (Sigma Aldrich) and Naphthol AS-MX phosphate (Sigma Aldrich). Images of plaques were taken with Axio microscope (Zeiss) using $2.5 \times$ lens. The plaques were outlined, 15 plaques were measured for each experiment, and the area of plaque were measured using ImageJ. Experiments were performed in triplicate.

### FC-TEM of BAC transfected cells

Monolayers of MeWo cells ($1.6 \times 10^5$ cell/cm$^2$) were seeded onto P100 dish and were transfected with 29 μg pOka-TKGFP, pOka-TKGFP-gB[E526P], pOka-TKGFP-gB[H527P], or pOka-TKGFP-gB[V528P] BACs by Lipofectamine 2000. Cells were harvested at 48 hrs post transfection, and resuspended in FACS staining buffer [2.5% FBS, 2 mM EDTA in PBS] at $5 \times 10^6$ cells/ml. The GFP-positive cells were sorted for enrichment by negative gating at 0.5% of the mock sample that was MeWo cells transfected without BAC. The sorted cells were pelleted and resuspended in 10% Gelatin in 0.1 M Sodium Cacodylate buffer pH 7.4 at 37 °C and allowed to equilibrate 5 min. Cells were pelleted again, excess gelatin removed, then chilled in cold blocks and covered with cold 1% Osmium tetroxide (EMS Cat# 19100) for 2 hrs rotating in a cold room, washed 3 times with cold ultrafiltered water, then en bloc stained overnight in 1% Uranyl Acetate at 4 °C while rotating. Samples were then dehydrated in a series of ethanol washes for 20 minutes each at 4 °C beginning at 30%, 50%, 70%, 95% where the samples were then allowed to rise to RT, changed to 100% ethanol, then Propylene Oxide (PO) for 15 min. Samples were infiltrated with EMbed-812 resin (EMS Cat#14120) mixed 1:2, 1:1, and 2:1 with PO, and then were placed into neat EMbed-812 for 2 to 4 hrs followed by placement into molds at 65 °C oven overnight. Sections were taken ~80 nm, picked up on formvar/Carbon coated 100 mesh Cu grids, and stained for 30 sec in 3.5% Uranyl Acetate in 50% Acetone followed by staining in 0.2% Lead Citrate for 3 min. Observed in the JEOL JEM-1400 120 kV and photos were taken using a Gatan Orius 4k × 4k digital camera. The yield efficiency of FC-TEM was summarized in Supplementary Table 4.

### Reporting summary

Further information on research design is available in the Nature Portfolio Reporting Summary linked to this article.

## Data availability

Data generated and/or analyzed during the current study are available in the paper or are appended as supplementary data, and data that support this study are available from the authors upon request. Cryo-ET maps have been deposited in the Electron Microscopy Data Bank (EMDB) with accession codes EMD-42250 (gB[H527P] class I) and EMD-42251 (gB[H527P] class II). The source data underlying Figs. 3c−e, 5b, S2b, S2d, S2e, S4c have been provided as a Source Data file. Source

data are provided with this paper. All primary data will be provided by the corresponding author upon request. Source data are provided with this paper.

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

## Acknowledgements

We thank the National Institutes of Health (NIH) for their valuable support through grants R01-AI102546 to A.M.A. and S.L.O. and R21-MH125285 to M.C. We thank the Stanford Cell Sciences Imaging Facility for supporting the confocal microscopy and the transmission electron microscopy (TEM) work, the project described was supported, in part, by Award Number 1S10OD010580-01A1 from the National Center for Research Resources (NCRR) (confocal microscopy work), and by NIH S10 Award Number 1S10OD028536-01, titled "OneView 4kX4k sCMOS camera for transmission electron microscopy applications" from the Office of Research Infrastructure Programs (ORIP) (TEM work). Its contents are solely the responsibility of the authors and do not necessarily represent the official views of the NCRR or the NIH. Cell sorting was performed in the Stanford Shared FACS Facility on an instrument obtained using NIH S10 Shared Instrument Grant S10RR025518-01. Molecular graphics and analyses performed with UCSF ChimeraX, developed by the Resource for Biocomputing, Visualization, and Informatics at the University of California, San Francisco, with support from NIH R01-GM129325 and the Office of Cyber Infrastructure and Computational Biology, National Institute of Allergy and Infectious Diseases. NAMD was developed by the Theoretical and Computational Biophysics Group in the Beckman Institute for Advanced Science and Technology at the University of Illinois at Urbana-Champaign. We gratefully acknowledge funding from the DFG [a Cluster of Excellence RESIST (EXC 2155) project and the BMBF (05K18BHA) to K.G. and a Marie Curie Individual Fellowship (HSV1EN-TRYPROTEINMAP) to E.M. Part of this work was performed at the Cryo-EM Facility at CSSB, supported by the UHH and DFG grant numbers (INST 152/772-1|152/774-1|152/775-1|152/776-1).

## Author contributions

M.Z. constructed mutant expressing plasmids, BAC recombinant viruses, carried out sequence alignments, flow cytometry, stable reporter fusion assays, plaque assays, immunoprecipitation, western blotting, confocal microscopy, and FC-TEM. B.V. and E.M. generated EVs, prepared Cryo-ET grids, carried out Cryo-ET data acquisition. M.C. performed subtomogram averaging analysis on the Cryo-ET data. S.L.O. conducted the AlphaFold2 simulation and performed model building, together with M.C.. A.M.A., K.G., S.L.O., B.V., and M.Z., conceptualized the project. All the authors, including W.C. analyzed and discussed the data and contributed intellectually to the manuscript.

## Competing interests

K.G. and B.V. are inventors on the U.S. patent application 17/484,583 and international patent application PCT/EP2022/076579, "Prefusion-Stabilized Herpesvirus Glycoprotein B". The remaining authors declare that they have no competing interests.
