## [Peer Review File · Nature Communications]

Caught in the Act: targeted mutagenesis of the herpesvirus fusogen central helix captures transition statesReviewers' Comments:

Reviewer #1:

Remarks to the Author:

gH-gL and gB are essential components of the entry machinery that is conserved in all herpesviruses. gH-gL triggers the viral fusogen gB to mediate fusion; however, the details of the structural changes in the gB trimer that mediate fusion are unresolved. This work begins to address this critical gap in knowledge, using cryoET to visualize VZV gB in a prefusion/intermediate state. Using a mutant form of gB, (gB[H527P]), two conformations (closed/class I and open/class II) of prefusion gB are visualized and the closed form represents a novel conformation. A comparison of the open and closed conformation reveals shifting of the fusion loops and rotation of domains II and III. The cellular processing and incorporation into virions are assessed for the gB[H527P] mutant as well as two other gB mutants.

The work is well contextualized, building on previous studies of HCMV and HSV gB. The data are obtained with appropriate techniques, presented in detail, and interpreted carefully. Supplemental data are well selected. The conclusions are supported by the evidence presented and the manuscript is well written. The work moves the field of herpesvirus entry forward, providing a snapshot of multiple forms of VZV gB prior to fusion. The work contributes to a mechanistic explanation of herpesvirus entry that is relevant to all herpesviruses and to other viral fusogens. These findings will serve as the basis for future studies, potentially including vaccine design, and the work will be widely cited.

Suggestions:

Fig. 2 and discussion. The authors could note where H527P lies in the gB class I and class II prefusion structures. The results suggest that H527P impacts the structure of prefusion form, so knowing the general location of the mutation would be useful.

Fig S2B, lower panel. The authors could comment on the ratio of immature uncleaved to cleaved gB, comparing WT gB to the mutants.

Line 128, Fig 2A. The authors state that gB clusters have "extensive edge contacts". Do the images provide enough detail to indicate which parts of the gB trimers participate in the contacts?

The authors could comment on whether class I and II may represent conformations of gB that are occur sequentially during fusion. Based on the HCMV gB model, would the authors propose that the class I structure transitions into the class II structure during entry? I do not want to encourage the authors to over-interpret the data (they did a nice job avoiding that), but I am curious about their expert opinions.

Line 109. This sentence states that EHV is in the "upper" region of the DIII central helix in the postfusion structure. They do not look like they are in the upper region of the helix in Fig. 1C. Clarification of what the authors intended here would be helpful.

Line 151. The authors state that the two structures provide evidence for the first transitional steps from a prefusion to postfusion form. Given that gB[H527P] mutant is a nonfunctional mutant, at least one of these structures could be an aberrant form. I suggest softening this conclusion by adding "potential" or "may" to the sentence.

Line 162. The authors note that mAb 93K must recognize an epitope on the prefusion form of gB because it is a neutralizing antibody. This is true; however, the 93K epitope could also be present on intermediate, postfusion, or aberrant forms of gB. The authors could acknowledge this caveat so that readers don't mistake 93K binding as conclusive evidence of the prefusion conformation. If additional

evidence exists to bolster the argument that 93K only recognizes prefusion gB, that evidence could be added here instead.

Fig 3C and Line 175. If mAb 93K recognizes the prefusion form of gB and gB[H527P] is locked in the prefusion conformation, why isn't binding of 93K to cells expressing gB[H527P] enhanced compared to WT gB?

Fig 3A, line 190. The alphafold modeling was performed on the DIII central helix in isolation. Although the DIII is extended in the postfusion conformation, the DIII helix is broken in the prefusion conformation. The alphafold prediction may not be relevant to the prefusion conformation. The proline substitutions used in this study appear to affect the gB structure prior to the extension of DIII.

Line 291. The meaning of "highly metastable" isn't completely clear. From context, the authors mean the metastable form is very unstable, but they could consider clarifying the phrasing.

Line 290. The authors state that WT HSV-1 gB shows a "partial transition" to the postfusion conformation. Upon initial reading, I thought this statement referred to WT HSV-1 gB trimers adopting an intermediate conformation. The authors could rephrase to indicate that a portion of the HSV-1 gB population adopts a postfusion conformation, whereas all of the VZV gB population adopts postfusion.

Line 299. The word 'The' starts with an odd font.

Line 294. The authors cite one PRV and two HSV-1 papers when they discuss a model that VZV gH-gL keeps gB "locked in place". These three papers examine how gH-gL interacts with gB prior to fusion, but they do not provide evidence that gH-gL is required to stabilize gB.

Line 568. Approximately how many plaques were measured? The statistics show significance, but a ballpark figure here would be nice.

Fig S2. Is the His-tag at the N-terminus of gB? Adding this to the figure legend would eliminate the need to find the answer.

Reviewer #2:

Remarks to the Author:

Review for NCOMMS-22-51317-T

In the manuscript "Caught in the Act: Targeted mutagenesis of the herpesvirus fusogen central helix captures transition states" by Zhou et al, the authors present a combination of approaches to characterize the prefusion state of the herpesvirus gB protein using extracellular vesicles and cryo-electron tomography. In general, Herpesviridae are important human pathogens and understanding their biology in depth is important for rational drug design and therapeutics. These capsids are difficult to study in their true native form as there is a complex tegument layer, and lipid surrounding their large, T=16 icosahedral capsid. The choice of cryo-tomography is a great approach to look at these membrane embedded proteins and understand fusion of these virions. Structure-guided mutagenesis approaches were also employed and reveal a single residue (V528) to be important for the function, and cell sorting assays are used to assay fusion. This work builds on previous knowledge gained by this group and others to provide a more thorough view of herpesvirus entry.

Unfortunately, there are several issues with the manuscript in its current state. The structures themselves are not of sufficient quality to support the authors conclusions and the writing of the manuscript is very obtuse. The jargon-heavy nature of the paper is not accessible to a broad audience and seems tailored more to a Herpes-specific readership. Some specific examples are listed below, but

in this reviewer's opinion this work is not ready for publication.

Major issues

1. The cryo-ET structures are of low resolution and somewhat questionable quality (the FSC curves for example display highly unusual features for a well defined map and the figure presented shows a rather dramatic overinterpretation of the maps). With the current data, there is very low confidence for docking the atomic models into these densities.
2. There is a lot of jargon and the Herpes naming conventions make the writing inaccessible to a broad audience. Perhaps a well labeled cartoon schematic or model with the key proteins featured would be helpful to the introduction.
3. The discussion could be slimmed excessively. As one example, the authors spend a lot of writing discussing each mutant made. It would be much more streamlined to say "only mutant V528P showed significant change" instead of describing all of them.
4. The alpha fold work is nice, but MD simulations or taking one step further might help support the relatively weak cryo-ET density.
5. Figure 5 is nice work, but better labels in panel B would help the reader interpret the bands better.

Reviewer #3:

Remarks to the Author:

In this study, the Zhou et al. investigate the role of the herpesvirus VZV gB DIII helix in membrane fusion by site-directed mutagenesis and cryo-electron tomography. The authors introduced a set of proline mutations in the DIII domain intended to destabilize helix formation that occurs during the conformational transition between prefusion and postfusion forms. The aim of doing this was to bias gB conformation towards the prefusion state and enable higher-resolution structure determination of this herpesvirus fusion protein as well as to test the role of DIII in gB biosynthesis and fusion activity. Using cryo-electron tomography, they show that the gB [H527P] mutation stabilizes an apparent prefusion conformation of the protein on the surface of extracellular vesicles. Using flow cytometry, they demonstrate the mutant binds mAb 93k, an antibody thought to be specific for the gB prefusion conformation.

Furthermore, using fusion and plaque assays, the authors show that disruption of the gB DIII helix is highly detrimental to fusion and infectivity of the viral particles. Analogous alanine mutants of the same residues, which are not expected to disrupt helix formation, only show slight decreases in fusogenicity and infectivity, supporting the conclusion that it is helix disruption and not modification of the individual 526-528 residues that severely impact the activity of the protein. Using confocal microscopy and western blotting, they show that for two of the three mutants tested (E526P and H527P), proper biosynthesis and trafficking of the gB protein is retained. Therefore, aside from mutant V528P, differences in fusogenicity are result of the effect of the mutation on the activity of the protein and not its cellular localization. The authors also utilize conventional electron microscopy to demonstrate the mutants do not interfere with VZV viral particle assembly.

Furthermore, using subtomogram averaging, the authors investigate the structure of gB [H527P] in the membranes of extracellular vesicles. Intriguingly, they show that two classes of prefusion-esque assemblies can be parsed from the tomographic images of the protein exist on the vesicle surface, which they interpret as possible early intermediates along the transition between pre- and post-fusion conformations. While this is a possibility, crucially, evidence is lacking to support the conclusion that the structures represent fusion transition states rather than an equilibrium of two alternate conformations or even possible artifacts of the gB over-expression in isolation rather than in complex with gH and gL viral proteins. The structures presented do represent a step forward in resolution of this class III fusion protein in the context of the membrane bilayer, but further modeling and detailed structural information is needed to conclusively identify the relation of the new structures to the gB-mediated fusion process.

General comments:

-The authors state (e.g. line 150) that the data provide evidence for the first transitional steps from pre- to post-fusion conformation, which seems highly speculative. I do not see evidence that these structures necessarily represent intermediates along the transition to post-fusion conformation, vs representing some conformations of gB close to its prefusion form that may or may not be on pathway or may be a result of the proline introduced into the DIII helix. This is the essential point and central conclusion of the study, thus more definitive evidence is needed to associate the structures with the pathway of conformational change gB takes during fusion.

-The visualization of two distinct classes of VZV gB [H527P] particles on EVs is quite interesting (Fig 2D-F). However, classification of subtomograms can sometimes result in separation based on the missing wedge artifact rather than actual conformational differences. In order to further support the conclusion that the two classes are distinct conformations, it would be helpful for the authors to provide plots of the angular distributions of particles from each class to demonstrate a full angular sampling is present for both classes. Additionally, it is not noted in the manuscript what percentage of subtomograms belong to each class for the VZV gB[H527P] prefusion structure (Fig 2D-F). It would be informative to provide this information. Does the molecule sample both states equally or is one state dominant?

-Alternatively, due to the high density of gB on EV surface resulting from over-expression of the protein, it seems conceivable that the two conformations may be related to the different local inter-spike environment and interactions rather than conformational states intrinsic to a gB trimer. The additional density extending laterally in class II may indicate an influence of surface density and inter-spike interactions in giving rise to the observed conformational classes. Specifically, on line 128, the authors mention that gB [H527P] clusters are evident with extensive edge contacts between molecules. Is there any evidence that these intermolecular contacts are specific and form a regular pattern, or is it possible they are nonspecific contacts due to the high expression of gB on EVs?

-In line 146, the authors describe several differences in the two gB [H527P] prefusion conformations and mention movements of specific domains that could account for the differences in electron density. The domain movements described may be plausible but would be far more convincingly demonstrated by flexibly fitting the VZV gB homology model into the individual maps. In the absence of more detailed structural modeling, it is difficult at the resolution attained for the EM maps to draw specific conclusions about loop and helix movements and positioning.

-It is generally believed that the additional gH and gL VZV glycoproteins play a key role in modulating gB activity, thus in the absence of these key proteins, it seems difficult to draw definitive conclusions about the nature of gB's prefusion conformation as it would exist on virions, let alone as relates to fusion intermediates.

-The application of Alphafold2 to modeling the DIII helix in isolation of tertiary and quaternary contacts is questionable. Yes, it may inform the impact of proline substitution in the helix (and proline's helix disrupting role has been long appreciated) but it is not equivalent to demonstrating the role of the mutations in the context of the gB trimer.

-The authors claim that their FC-TEM assay is "novel" (e.g. line 96). However, while these exact methods have not been applied before, the use of flow-cytometry to separate cells or cellular components prior to electron microscopy is not a novel concept and has been used by several other groups previously.

-Based on similar binding of mAb 93K binding to H527P to WT gB, the authors state that the mutant gB has adopted a prefusion conformation: "93k mAb binding to gB[H527P] was comparable to WT gB

(Fig. 3C), indicating preservation of the conformational 93k epitope and confirming that the gB[H527P] cryo-EM structures represent forms of gB locked in prefusion conformations.” However it is not demonstrated which of the two conformations (or both) are involved in this binding response. Structural characteriation of the antibody Fab in complex with gB H527P on EV would provide much more tangible evidence to link the structures with the apparent antibody binding behavior. Alternatively, like many fusion proteins, if gB undergoes breathing motions and transient exposure of epitopes that 93k binds, it may interact with a conformation not even represented by either class I or II reconstructions.

Minor points:

Figure 1, seems like Figures B and C positions, showing pre vs post fusion, could be switched to harmonize with the tomograms shown in panel D

In line 862 the authors inaccurately refer to their method as “negative staining”. While the application of uranyl acetate to proteins/samples on an EM grid resulting in fixation of samples in a heavy metal crust over the sample providing inverted contrast and thus is termed negative staining, the application of heavy metal stains such as uranyl acetate and osmium tetroxide onto resin embedded cells results in interaction of the heavy metals with charged lipids, proteins, and nucleic acids results in positive staining of the cellular structures formed by these macromolecules at the resolution provided by the technique.

REVIEWER COMMENTS

Reviewer #1 (Remarks to the Author):

gH-gL and gB are essential components of the entry machinery that is conserved in all herpesviruses. gH-gL triggers the viral fusogen gB to mediate fusion; however, the details of the structural changes in the gB trimer that mediate fusion are unresolved. This work begins to address this critical gap in knowledge, using cryoET to visualize VZV gB in a prefusion/intermediate state. Using a mutant form of gB, (gB[H527P]), two conformations (closed/class I and open/class II) of prefusion gB are visualized and the closed form represents a novel conformation. A comparison of the open and closed conformation reveals shifting of the fusion loops and rotation of domains II and III. The cellular processing and incorporation into virions are assessed for the gB[H527P] mutant as well as two other gB mutants.

The work is well contextualized, building on previous studies of HCMV and HSV gB. The data are obtained with appropriate techniques, presented in detail, and interpreted carefully. Supplemental data are well selected. The conclusions are supported by the evidence presented and the manuscript is well written. The work moves the field of herpesvirus entry forward, providing a snapshot of multiple forms of VZV gB prior to fusion. The work contributes to a mechanistic explanation of herpesvirus entry that is relevant to all herpesviruses and to other viral fusogens. These findings will serve as the basis for future studies, potentially including vaccine design, and the work will be widely cited.

Author Reply: Thank you! We appreciate this positive assessment of our work.

Suggestions:

Fig. 2 and discussion. The authors could note where H527P lies in the gB class I and class II prefusion structures. The results suggest that H527P impacts the structure of prefusion form, so knowing the general location of the mutation would be useful.

Author Reply: Thank you for this suggestion. The ⁵²⁶EHV⁵²⁸ residues are predicted to be located in the upper region of the central helix in the prefusion structure based on the homology model as shown in Fig. 1b. We have now provided the location of H527P in the class I and class II prefusion structures in a new figure, Fig. S3a. As expected, a bend in the upper region of the central helix is shared by the two structures as shown in Fig. 2e and 2f, which is consistent with the published HSV-1 gB[H516P] prefusion structure, likely due to the proline substitution of this histidine residue in the central helix.

Fig S2B, lower panel. The authors could comment on the ratio of immature uncleaved to cleaved gB, comparing WT gB to the mutants.

Author Reply: This is a good suggestion, we have added the ratio of the immature uncleaved to mature cleaved gB for WT and all the mutant gB in Fig S2b.

Line 128, Fig 2A. The authors state that gB clusters have "extensive edge contacts". Do the images provide enough detail to indicate which parts of the gB trimers participate in the contacts?

Author Reply: Yes, for some regions, there were edge contacts between gB molecules on the surface of the extracellular vesicles as seen in Fig. 2a. Therefore, segmentation was performed to mask out the neighboring contacts at the edges, leaving the central region of the cryo-ET as seen in Fig. 2d and Fig. 2e. As outlined in the revised manuscript, MDFF has now been performed to fit the VZV gB homology model into the cryo-ET maps (Line 146-155, Fig. 2e and 2f and Movie. S1). Although the map resolutions do not provide enough detail to define the contacts between the gB domains, we speculate that DI and potential glycosylation could be involved because DI sits at the outmost edge of the structures for both gB[H527P] classes.

The authors could comment on whether class I and II may represent conformations of gB that are occur sequentially during fusion. Based on the HCMV gB model, would the authors propose that the class I structure transitions into the class II structure during entry? I do not want to encourage the authors to over-interpret the data (they did a nice job avoiding that), but I am curious about their expert opinions.

Author Reply: Thank you for inquiring about our opinion on this very curious question. Unfortunately, the lack of temporal information in our data cloud our ability to derive the sequential events underlying the transition of herpesvirus gB fusion into different conformations. However, the new information in our manuscript do provide, for the first time, a glimpse at herpesvirus glycoprotein dynamics in situ. To address your question, while the overall conformation is similar between HSV-1 and HCMV prefusion gB, HSV-1 prefusion gB has a bend in the central helix DIII whereas DIII in HCMV prefusion gB is straight. This structural variability is likely due to the different methodology used to stabilize the prefusion gB; a point mutation H516P for HSV-1 gB, and the combined fusion inhibitor and cross linking for HCMV gB. As the reviewer noted, we have been cautious in our speculation about what the two classes of VZV gB[H527P] reveal. It is important to note that WT VZV gB expressed on the extracellular vesicles (EVs) were exclusively, in our dataset, in the elongated postfusion form (Fig. 1d, Line 118-119). This contrasts with WT HSV-1 gB, where only 30% of the trimer were found in the elongated postfusion form, and 70% are in the compact prefusion form^{1,2}. Thus, VZV gB has a higher propensity for conformational change to the postfusion structure, indicating that VZV prefusion gB is energetically less stable than HSV-1 prefusion gB. Due to this property of VZV gB, we postulate that H527P creates two kinetic traps for VZV gB, class I and class II. Therefore, we speculate that the two classes of VZV prefusion gB discovered in this manuscript likely represent the prefusion structure or intermediate structures that tried to overcome the structural restrain exerted by the proline substitution but failed, essentially stalled in action, and likely ‘bouncing’ between two energetically favorable prefusion conformations. Therefore, we conclude that additional gB prefusion conformations can populate herpesvirus envelope. Consistent with this concept, the authors that solved the HCMV prefusion gB structure stated that “...the prefusion structure we have determined representing a “snapshot of a “breathing” molecule...” in their recent patent³, suggesting the published structure of prefusion HCMV gB might also represent an intermediate conformation. We have revised Fig. 2e and 2f and Movie. S1 to show the simulation of the molecular movement between the two classes of VZV gB[H527P] by MDFF, which might show the potential “breathing” between the two structures. However, because VZV and HSV-1 gB were locked in the prefusion conformation using a point mutation rather than the chemical approaches used for HCMV, we cannot use HCMV prefusion gB as a model to speculate whether class I transitions to class II or vice versa.

Line 109. This sentence states that EHV is in the "upper" region of the DIII central helix in the postfusion structure. They do not look like they are in the upper region of the helix in Fig. 1C. Clarification of what the authors intended here would be helpful.

Author Reply: Thank you for pointing this out. We understand that the confusion comes from the orientation of the prefusion and postfusion molecules relative to the membrane. We have changed the sentence accordingly in Line 101-104.

Line 151. The authors state that the two structures provide evidence for the first transitional steps from a prefusion to postfusion form. Given that gB[H527P] mutant is a nonfunctional mutant, at least one of these structures could be an aberrant form. I suggest softening this conclusion by adding "potential" or "may" to the sentence.

Author Reply: Thank you for this comment, we softened the conclusion accordingly in Line 154-155. However, we don't agree on the statement that "Given that gB[H527P] mutant is a nonfunctional mutant, at least one of these structures could be an aberrant form". We interpret "an aberrant form" to mean a seriously misfolded protein, which is not the case for gB[H527P] because we demonstrated that the mutant protein underwent conventional biosynthesis and intracellular trafficking process as WT gB did, suggesting proper protein folding. In contrast, gB[V528P] in our study provides an example for "an aberrant form". This mutant was predominantly restricted to the ER and was not trafficked to cell surfaces or found in EVs likely due to lysosomal degradation as a result of protein misfolding. Furthermore, the neutralizing antibody 93k recognized gB[H527P], strongly suggesting that the epitope for 93k is preserved in the mutant protein even though the mutant protein was not able to complete the conformational change to postfusion structure. In addition, based on the HSV-1 gB[H516P] mutant, we are confident that the two classes represent intermediate transitional steps. X-ray crystallography of the soluble HSV-1 gB[H516P] ectodomain (expressed without cytoplasmic domain) showed that this mutant formed a postfusion structure in the absence of the cytoplasmic domain and the lipid bilayer (Vollmer et al, 2020¹, Fig. S2). As VZV gB is energetically less stable than HSV-1 gB, we predict that VZV gB[H527P] would also preferentially adopt the postfusion conformation when expressed as a soluble ectodomain. As outlined above, our data support that class I and class II VZV gB[H527P] represent two kinetically favorable conformations that have the potential to transition into a postfusion conformation if the mutant was not constrained by the membrane and cytoplasmic domain.

Line 162. The authors note that mAb 93K must recognize an epitope on the prefusion form of gB because it is a neutralizing antibody. This is true; however, the 93K epitope could also be present on intermediate, postfusion, or aberrant forms of gB. The authors could acknowledge this caveat so that readers don't mistake 93K binding as conclusive evidence of the prefusion conformation. If additional evidence exists to bolster the argument that 93K only recognizes prefusion gB, that evidence could be added here instead.

Author Reply: We appreciate this point and have revised the manuscript accordingly. We agree that just because an antibody is neutralizing doesn't mean it can only bind to the prefusion structure. mAb 93k's conformational epitope was mapped to DIV of postfusion VZV gB in a 2.8 Å 93k-gB complex by our lab previously⁴. This antibody also prevents VZV infection by inhibiting gB fusion⁴. Due to the energetically unstable nature of VZV gB, the likelihood of 93k

to bind an intermediate conformation is unlikely. Thus, mAb 93k very likely binds to both prefusion and postfusion VZV gB. Text has been revised at Line 163-168.

Although binding of mAb 93k to an aberrant form of gB cannot be ruled out, it does detect gB[H527P] to comparable levels as WT gB (Fig. 3b and 3c and Fig. S4b), indicating that the epitope of 93k is intact in gB[H527P], presumably in both classes. Importantly, mAb 93k binding to the misfolded gB[V528P] was largely limited compared to gB[WT] and gB[H527P] (Fig. S4b), suggesting that binding to aberrant forms of gB could be limited. We have added a new supplementary figure Fig. S3b to visually demonstrate where 93k Fab binds to the gB[H527P] class I and class II MDFF-based structures. DIV is inverted in the prefusion conformation of gB compared to the postfusion form. DIV of the gB-93k structure (PDB number 6VN1) was aligned with DIV of the prefusion model, placing the Fab fragments of 93k on the class I and II models. A corresponding sentence was added in Line 183-185.

Taken together, the binding of 93k neutralizing antibody to gB[H527P] was used as a test for the preservation of its conformational epitope in DIV of gB, supporting the prefusion nature of this mutant protein.

Fig 3C and Line 175. If mAb 93K recognizes the prefusion form of gB and gB[H527P] is locked in the prefusion conformation, why isn't binding of 93K to cells expressing gB[H527P] enhanced compared to WT gB?

Author Reply: Thank you for this question. This point was also raised by reviewer #3 and is a concept we've rationalized as follows: as aforementioned, mAb 93k neutralizes VZV by preventing gB/gH-gL-mediated fusion and was shown to bind DIV of postfusion gB by near-atomic resolution (2.8Å) cryo-EM⁴. Critically, the 93k epitope is accessible in the prefusion conformation of VZV gB based on the homology modelling with HSV-1 prefusion gB⁵. As gB is a trimer, there are three epitopes, which are all accessible in the prefusion and postfusion states of gB. The 93k epitope also remains accessible for the two classes of gB[H527P] (see the newly added Fig. S3b, Line 183-185). In addition, WT VZV gB is predominantly presented as a postfusion conformation on EV membranes (add Line 118-119). Based on our previous studies and new data presented in this manuscript (Fig. 3b, 3c, and S3b), it is unsurprising that 93k can bind equally well to prefusion gB (gB[H527P]) and postfusion gB (WT gB).

Fig 3A, line 190. The alphafold modeling was performed on the DIII central helix in isolation. Although the DIII is extended in the postfusion conformation, the DIII helix is broken in the prefusion conformation. The alphafold prediction may not be relevant to the prefusion conformation. The proline substitutions used in this study appear to affect the gB structure prior to the extension of DIII.

Author Reply: In the HCMV prefusion model of gB, the DIII helices are structurally similar to those in the postfusion conformation. In contrast, for HSV-1 gB[H516P] the DIII helices are slightly curved in the prefusion conformation, which we predicted to be attributed to the proline substitution. As prolines are known to disrupt helices, AlphaFold2 was used to explore whether the proline substitutions at ⁵²⁶EHV⁵²⁸ in VZV gB would induce a similar curvature. Although minor, E526P and H527P were predicted to induce curvature into the DIII helix (Fig. 3a), supporting the notion that the deformation prevented the canonical molecular transition of gB from a prefusion to postfusion conformation, evident by the loss of function of those two mutant proteins in fusion assay. Furthermore, this was dramatically illustrated for V528P where the DIII helix was predicted to bend at almost 90° when predicted as a monomer, which was consistent

with aberrant biosynthesis. Thus, we respectively disagree that the proline substitutions are not relevant to the prefusion structure; each of the proline mutations are either kinetically trapped (E526P and H527P) or misfolded (V528P).

Line 291. The meaning of "highly metastable" isn't completely clear. From context, the authors mean the metastable form is very unstable, but they could consider clarifying the phrasing.

Author Reply: We have now rearranged and revised that part of the discussion to address this and also your following questions, so please find the revised Line 291-312 regarding this point.

Line 290. The authors state that WT HSV-1 gB shows a "partial transition" to the postfusion conformation. Upon initial reading, I thought this statement referred to WT HSV-1 gB trimers adopting an intermediate conformation. The authors could rephrase to indicate that a portion of the HSV-1 gB population adopts a postfusion conformation, whereas all of the VZV gB population adopts postfusion.

Author Reply: Thanks! We revised the text in Line 305-310 accordingly to clarify this notion that WT VZV gB readily adopts postfusion conformation (Fig. 1d) while only 30% of WT HSV-1 gB is in postfusion conformation^{1,2}.

Line 299. The word 'The' starts with an odd font.

Author Reply: Thank you for identifying this formatting error.

Line 294. The authors cite one PRV and two HSV-1 papers when they discuss a model that VZV gH-gL keeps gB "locked in place". These three papers examine how gH-gL interacts with gB prior to fusion, but they do not provide evidence that gH-gL is required to stabilize gB.

Author Reply: Thank you for raising this important point. We have revised the information in the discussion, Line 296-312, for clarification. Although this is a very active area of research, the precise role that gH-gL plays in regulating gB fusion function remains elusive. What is clear is that for alpha-, beta- and gammaherpesviruses, gH-gL is required for gB to function in fusion. However, there are currently two additional hypotheses related to gH-gL regulation of gB: 1) gH-gL acts as activator to trigger gB conformational change by relaying the activation signal from either a receptor (EBV, HCMV, KSHV) or a gD conformational change in the case of HSV; 2) gH-gL acts as a fusion regulator preventing gB from premature activation. Mutagenesis studies, published by us and others support both hypotheses. Moreover, physical interactions between gB and gH-gL have been reported, intracellularly in ER and on virion envelope^{6,7}. How such interaction affects gB function and stabilize its prefusion conformation is unknown, although gB/gH-gL interaction can be detected before and during fusion⁸. However, the cytoplasmic domain (CTD) of gH is likely involved in gB fusion regulation since truncation of the tail of gH CTD resulted in increased gB fusion in both VZV and HSV⁹⁻¹¹.

Line 568. Approximately how many plaques were measured? The statistics show significance, but a ballpark figure here would be nice.

Author Reply: For each experiment, three replicates were performed with 15 plaques measured for each replicate. The statistical analysis was provided in source data for Fig. 2e. This information has been added to Line 568 in the Methods and the figure legend of Fig. 2e

Fig S2. Is the His-tag at the N-terminus of gB? Adding this to the figure legend would eliminate the need to find the answer.

Author Reply: The His-tag is at the C-terminus of gB. We added this information in the Methods, Line 438 and also in the Figure legend of Fig. S2c.

Reviewer #2 (Remarks to the Author):

Review for NCOMMS-22-51317-T

In the manuscript “Caught in the Act: Targeted mutagenesis of the herpesvirus fusogen central helix captures transition states” by Zhou et al, the authors present a combination of approaches to characterize the prefusion state of the herpesvirus gB protein using extracellular vesicles and cryo-electron tomography. In general, Herpesviridae are important human pathogens and understanding their biology in depth is important for rational drug design and therapeutics. These capsids are difficult to study in their true native form as there is a complex tegument layer, and lipid surrounding their large, T=16 icosahedral capsid. The choice of cryo-tomography is a great approach to look at these membrane embedded proteins and understand fusion of these virions. Structure-guided mutagenesis approaches were also employed and reveal a single residue (V528) to be important for the function, and cell sorting assays are used to assay fusion. This work builds on previous knowledge gained by this group and others to provide a more thorough view of herpesvirus entry.

Author Reply: Thank you for the positive evaluation of our work.

Unfortunately, there are several issues with the manuscript in its current state. The structures themselves are not of sufficient quality to support the authors conclusions and the writing of the manuscript is very obtuse. The jargon-heavy nature of the paper is not accessible to a broad audience and seems tailored more to a Herpes-specific readership. Some specific examples are listed below, but in this reviewer’s opinion this work is not ready for publication.

Author reply: Thank you for your suggestions. We have made revision to address your concerns and questions below.

Major issues

1. The cryo-ET structures are of low resolution and somewhat questionable quality (the FSC curves for example display highly unusual features for a well defined map and the figure presented shows a rather dramatic overinterpretation of the maps). With the current data, there is very low confidence for docking the atomic models into these densities.

Author Reply: Thank you for your concerns on the current resolution of the cryo-ET structures. To address this, we have provided further metrics that support our cryo-ET reconstructions in the revised Fig. S2 and text in result Line 128-155 and in discussion Line 313-338. The unusual features displayed in the FSC curve (Fig. 2c) was determined to be caused by the heterogeneous nature of the VZV gB[H527P] population on EV membranes as shown in Fig. S2f. From the 2074 particles of gB[H527P], three classes were originally identified each with random distribution (Fig. S2f-S2h), but class III was discarded as the central region of this class was poorly resolved, indicating the proposed movement of DII and DIII between class I and class II

towards the center of the structure (revised Fig. 2f, and Movie. S1). As a result, classification improved the FSC curves, generating the class I (9.8Å) and class II (12.4Å) cryo-ET maps. We agree that fitting atomic models of VZV gB into the class I and class II maps was not possible, hence our original figures overlaid the VZV gB homology model on the class I and class II cryo-ET maps. As outlined in our response to reviewer #1, we have addressed the fitment of the model by using MDFF to lock and fit the domains of the VZV gB prefusion model into the two classes (Fig. 2e; Movie. S1). This now visually demonstrates the motion of DI and DIII as the VZV gB[H527P] undergoes a transition between the two kinetically trapped conformations. In our response to reviewer #1, we note that WT VZV gB expressed on the EVs are predominantly in a postfusion conformation (Fig. 1d), whereas only 30% of the WT HSV-1 gB are found in this conformation^{1,2}. This indicates that VZV gB has a lower energy barrier to undergo from prefusion to postfusion conformation. We postulate that this property of VZV gB explains the heterogeneity of VZV gB[H527P] and why two kinetically trapped classes were captured in the cryo-ET data.

2. There is a lot of jargon and the Herpes naming conventions make the writing inaccessible to a broad audience. Perhaps a well labeled cartoon schematic or model with the key proteins featured would be helpful to the introduction.

Author Reply: We agree that the herpesvirus naming conventions can be inaccessible. We did aim to make our work more accessible to a wider audience in the original manuscript. At the reviewer's request, we have addressed this comment further by additional revisions to the text.

3. The discussion could be slimmed excessively. As one example, the authors spend a lot of writing discussing each mutant made. It would be much more streamlined to say "only mutant V528P showed significant change" instead of describing all of them.

Author Reply: The reason we went through each of the mutant was because proline substitution at each of those positions had different phenotype on gB mediated fusion, gB biosynthesis, and virus replication. The structural impact on the central helix and functional impact on gB fusion seems to be closely related to where the proline was introduced. Therefore, we respectively disagree "only mutant V528P showed significant change" and think a discussion on each of the mutant is important for readers to understand the structure and function study we provide in this paper.

4. The alpha fold work is nice, but MD simulations or taking one step further might help support the relatively weak cryo-ET density.

Author Reply: Thank you for the appreciation on our prediction work done using AlphaFold2. As suggested, we now included MDFF to predict potential movement between the two classes of gB[H527P] prefusion structures (revised Fig.2e-2f, and Movie.S1).

5. Figure 5 is nice work, but better labels in panel B would help the reader interpret the bands better.

Author Reply: Thank you for the positive feedback for this Figure! We have revised the labels in Fig. 5b as suggested.

Reviewer #3 (Remarks to the Author):

In this study, the Zhou et al. investigate the role of the herpesvirus VZV gB DIII helix in membrane fusion by site-directed mutagenesis and cryo-electron tomography. The authors introduced a set of proline mutations in the DIII domain intended to destabilize helix formation that occurs during the conformational transition between prefusion and postfusion forms. The aim of doing this was to bias gB conformation towards the prefusion state and enable higher-resolution structure determination of this herpesvirus fusion protein as well as to test the role of DIII in gB biosynthesis and fusion activity. Using cryo-electron tomography, they show that the gB [H527P] mutation stabilizes an apparent prefusion conformation of the protein on the surface of extracellular vesicles. Using flow cytometry, they demonstrate the mutant binds mAb 93k, an antibody thought to be specific for the gB prefusion conformation.

Furthermore, using fusion and plaque assays, the authors show that disruption of the gB DIII helix is highly detrimental to fusion and infectivity of the viral particles. Analogous alanine mutants of the same residues, which are not expected to disrupt helix formation, only show slight decreases in fusogenicity and infectivity, supporting the conclusion that it is helix disruption and not modification of the individual 526-528 residues that severely impact the activity of the protein. Using confocal microscopy and western blotting, they show that for two of the three mutants tested (E526P and H527P), proper biosynthesis and trafficking of the gB protein is retained. Therefore, aside from mutant V528P, differences in fusogenicity are result of the effect of the mutation on the activity of the protein and not its cellular localization. The authors also utilize conventional electron microscopy to demonstrate the mutants do not interfere with VZV viral particle assembly.

Furthermore, using subtomogram averaging, the authors investigate the structure of gB [H527P] in the membranes of extracellular vesicles. Intriguingly, they show that two classes of prefusion-esque assemblies can be parsed from the tomographic images of the protein exist on the vesicle surface, which they interpret as possible early intermediates along the transition between pre- and post-fusion conformations. While this is a possibility, crucially, evidence is lacking to support the conclusion that the structures represent fusion transition states rather than an equilibrium of two alternate conformations or even possible artifacts of the gB over-expression in isolation rather than in complex with gH and gL viral proteins. The structures presented do represent a step forward in resolution of this class III fusion protein in the context of the membrane bilayer, but further modeling and detailed structural information is needed to conclusively identify the relation of the new structures to the gB-mediated fusion process.

Author Reply: Thank you for the positive comments on our work and also giving us insightful suggestions for revision.

General comments:

-The authors state (e.g. line 150) that the data provide evidence for the first transitional steps from pre- to post-fusion conformation, which seems highly speculative. I do not see evidence that these structures necessarily represent intermediates along the transition to post-fusion conformation, vs representing some conformations of gB close to its prefusion form that may or may not be on pathway or may be a result of the proline introduced into the DIII helix. This is the essential point and central conclusion of the study, thus more definitive evidence is needed to associate the structures with the pathway of conformational change gB takes during fusion.

Author Reply: Thank you for your assessment of our findings. The central hypothesis is that the VZV gB[H527P] mutation, like HSV-1 gB[H516P], locks gB in a prefusion conformation. Based on the cryo-ET maps for this and previous studies^{1,2,7}, VZV gB is locked in a prefusion conformation on EVs. Our advanced approach in STA and classification revealed three classes, class I and II with defined features that resemble prefusion gB and a third class with less defined features. As we outlined in our response to reviewer #1 and #2, there are three critical points that support our conclusion of identifying intermediate states. Firstly, HSV-1 gB[H516P], which is locked in a prefusion conformation on EVs, can transition to a postfusion conformation when expressed as a soluble form¹; VZV gB[H527P] will likely behave in a similar manner given the highly conserved nature of herpesvirus gB. Secondly, a characteristic of VZV gB is the reduced stability compared to HSV-1 gB because WT VZV gB predominantly adopts a postfusion conformation on EVs (Fig. 1d and Line 118-119), whereas only 30% of the WT HSV-1 gB are in the postfusion form^{1,2}. Thirdly, the low energy barrier of WT VZV gB to adopt a postfusion conformation on EVs indicates an energetically dynamic molecule, supported by the heterogeneity of VZV gB[H527P] on the EVs. Together, these new findings reveal gB dynamics in the context of a lipid bilayer, providing important information about domain movement immediately prior to the conformational change to a postfusion state. Such information will be valuable to the wider herpesvirus and virus fusion community for rationale vaccine design. We do acknowledge that our current data are unable to provide the necessary temporal information to determine which of the two classes are closest to the prefusion state, something we were careful to avoid in the original manuscript as we felt this would be over interpretation. To provide clarification, we have revised the Fig. 2, Fig. S2 and Movie. S1 and added corresponding sentences in the text in line 120-155 for clarification. In addition to our cryo-ET findings, we also provided considerable support via our structure-function analyses revealing that gB[H527P] was defective in gB/gH-gL mediated cell-cell fusion, but underwent canonical biogenesis, and preserved the conformational epitope of a human neutralizing antibody 93k. Conservation of the conformational epitope of the neutralizing antibody supported the prefusion nature of gB[H527P]. These data strongly support that gB[H527P], fusion incompetent, was trapped in prefusion states constrained by the proline substitution in the central helix.

-The visualization of two distinct classes of VZV gB [H527P] particles on EVs is quite interesting (Fig 2D-F). However, classification of subtomograms can sometimes result in separation based on the missing wedge artifact rather than actual conformational differences. In order to further support the conclusion that the two classes are distinct conformations, it would be helpful for the authors to provide plots of the angular distributions of particles from each class to demonstrate a full angular sampling is present for both classes. Additionally, it is not noted in the manuscript what percentage of subtomograms belong to each class for the VZV gB[H527P] prefusion structure (Fig 2D-F). It would be informative to provide this information. Does the molecule sample both states equally or is one state dominant?

Author Reply: Thank you very much for the advice on potential missing-wedge effects in our cryo-ET data. We have provided additional data as you suggested in Fig. S2f to show the particle picking scheme and the particle numbers belonging to each class, Fig. S2g to show the random distribution of each class on the membrane of EVs, and Fig. S2h to show the full angular sampling of the particles.

-Alternatively, due to the high density of gB on EV surface resulting from over-expression of the

protein, it seems conceivable that the two conformations may be related to the different local inter-spike environment and interactions rather than conformational states intrinsic to a gB trimer. The additional density extending laterally in class II may indicate an influence of surface density and inter-spike interactions in giving rise to the observed conformational classes. Specifically, on line 128, the authors mention that gB [H527P] clusters are evident with extensive edge contacts between molecules. Is there any evidence that these intermolecular contacts are specific and form a regular pattern, or is it possible they are nonspecific contacts due to the high expression of gB on EVs?

Author Reply: Thank you for raising this important question. This point was also raised by reviewer #1 and we have provided our response there, please review our response to reviewer #1 above. We also provided further clarification with the addition of Fig. S2g.

-In line 146, the authors describe several differences in the two gB [H527P] prefusion conformations and mention movements of specific domains that could account for the differences in electron density. The domain movements described may be plausible but would be far more convincingly demonstrated by flexibly fitting the VZV gB homology model into the individual maps. In the absence of more detailed structural modeling, it is difficult at the resolution attained for the EM maps to draw specific conclusions about loop and helix movements and positioning.

Author Reply: We agree with your assessment of the information provided in the original manuscript. To address this comment, we have performed MDFF of gB[H527P] domains I-V as outlined in our response to reviewer #1 (revised Fig. 2e and 2f, and revised Movie. S1).

-It is generally believed that the additional gH and gL VZV glycoproteins play a key role in modulating gB activity, thus in the absence of these key proteins, it seems difficult to draw definitive conclusions about the nature of gB's prefusion conformation as it would exist on virions, let alone as relates to fusion intermediates.

Author Reply: The gH-gL modulation hypothesis is central to herpesvirus membrane fusion, a subject also raised by reviewer #1; please review our response to reviewer #1 above. The critical aspect in our study is that gB has been kinetically locked by H527P in DIII central helix, which is hypothesized to occur in the presence of gH-gL during herpesvirus replication as shown by the inactivation of gB[H527P] mutant virus. Despite numerous publications on the topic of gH-gL interacting with and activating gB to trigger fusion, to date, the direct interaction of prefusion gB and gH-gL has only been visualized at low resolution on HCMV particles⁷. Prefusion HCMV gB on virus particles in isolation, without gH-gL, has also been visualized⁷. Critically, the gH-gL homodimer is not required to stabilize WT HSV-1 gB because, when expressed alone, the trimer readily forms a prefusion conformation on EVs^{1,2}. Recent work by Pataki et al. showed that the proline stabilized, fusion incompetent, prefusion HSV-1 gB[H516P] could bind gH-gL at similar levels compared to WT gB⁸, demonstrating that locked gB molecules can interact with gH-gL. Together, these data raise questions about the role of gH-gL in fusion activation rather than the nature of the kinetically locked prefusion structures adopted by VZV gB[H527P]. Importantly, studying gB in isolation on EVs provided the opportunity to reveal molecular characteristics of this trimer in the absence of additional constraints such as those imposed by gH-gL, for instance, revealing the heterogeneity of VZV prefusion gB[H527P] as shown in this manuscript.

-The application of Alphafold2 to modeling the DIII helix in isolation of tertiary and quaternary contacts is questionable. Yes, it may inform the impact of proline substitution in the helix (and proline's helix disrupting role has been long appreciated) but it is not equivalent to demonstrating the role of the mutations in the context of the gB trimer.

Author Reply: Thank you for raising this question about AlphaFold2. A related question was also raised by reviewer #1; please review our response above. We do agree that using AlphaFold2 on the DIII helix in isolation does not predict what would happen to the entire gB molecule, but it did provide an explanation for the misfolded gB[V528P] and provided supporting data for the effects of the gB[E526P] and gB[H527P] mutants. One important consideration for the application of Alphafold2 was that it performs very poorly with single point mutation within entire molecules. This has been our personal experience, including the proline substitutions at ⁵²⁶EHV⁵²⁸ in VZV gB, and has been recently reported by other group ¹², which is unsurprising given that AlphaFold2 is not expected to produce an unfolded protein structure given a sequence containing a destabilizing point mutation (<https://alphafold.ebi.ac.uk/fag>).

-The authors claim that their FC-TEM assay is “novel” (e.g. line 96). However, while these exact methods have not been applied before, the use of flow-cytometry to separate cells or cellular components prior to electron microscopy is not a novel concept and has been used by several other groups previously.

Author Reply: Thanks for this comment. As suggested, we removed the word “novel”.

-Based on similar binding of mAb 93K binding to H527P to WT gB, the authors state that the mutant gB has adopted a prefusion conformation: “93k mAb binding to gB[H527P] was comparable to WT gB (Fig. 3C), indicating preservation of the conformational 93k epitope and confirming that the gB[H527P] cryo-EM structures represent forms of gB locked in prefusion conformations.” However it is not demonstrated which of the two conformations (or both) are involved in this binding response. Structural characteriation of the antibody Fab in complex with gB H527P on EV would provide much more tangible evidence to link the structures with the apparent antibody binding behavior. Alternatively, like many fusion proteins, if gB undergoes breathing motions and transient exposure of epitopes that 93k binds, it may interact with a conformation not even represented by either class I or II reconstructions.

Author Reply: Thank you for this insightful comment. In our previous work we have mapped the epitope of 93k binding to DIV in a 2.8Å cryo-EM map of postfusion VZV gB ⁴. To address your question, 93k Fab fragment binding has been modelled for both the MDFF derived class I and class II gB[H527P] prefusion structures; the 93k epitope on DIV is accessible for class I and II. We have provided a new supplementary figure (Fig. S3b) to show this, and revised the manuscript text by adding lines 183-185.

Minor points:

Figure 1, seems like Figures B and C positions, showing pre vs post fusion, could be switched to harmonize with the tomograms shown in panel D

Author Reply: Good suggestion. The panels in Fig.1 have been revised accordingly.

In line 862 the authors inaccurately refer to their method as “negative staining”. While the application of uranyl acetate to proteins/samples on an EM grid resulting in fixation of samples

in a heavy metal crust over the sample providing inverted contrast and thus is termed negative staining, the application of heavy metal stains such as uranyl acetate and osmium tetroxide onto resin embedded cells results in interaction of the heavy metals with charged lipids, proteins, and nucleic acids results in positive staining of the cellular structures formed by these macromolecules at the resolution provided by the technique.

Author Reply: Thank you for giving us this suggestion, we have made changes accordingly in the Fig 4, and its figure legend.

References

- 1 Vollmer, B. *et al.* The prefusion structure of herpes simplex virus glycoprotein B. *Sci Adv* **6**, doi:10.1126/sciadv.abc1726 (2020).
- 2 Zeev-Ben-Mordehai, T. *et al.* Two distinct trimeric conformations of natively membrane-anchored full-length herpes simplex virus 1 glycoprotein B. *Proc Natl Acad Sci U S A* **113**, 4176-4181, doi:10.1073/pnas.1523234113 (2016).
- 3 Dormitzer, P. R. N., NY, US), Che, Ye (Niantic, CT, US), Chi, Xiaoyuan Sherry (Tenafly, NJ, US), Han, Seungil (Mystic, CT, US), Heim, Kyle Paul (Boulder, CO, US), Jones, Thomas Richard (Bluffton, SC, US), Liu, Yuhang (South Glastonbury, CT, US), Qiu, Xiayang (Mystic, CT, US), Yang, Xinzhen (Woodcliff, NJ, US), Yao, Xiaojie (East Lyme, CT, US), Griffor, Matthew Curtis (North Stonington, CT, US), Nicki, Jennifer Anne (Gales Ferry, CT, US). HUMAN CYTOMEGALOVIRUS GB POLYPEPTIDE. United States patent (2020).
- 4 Oliver, S. L. *et al.* A glycoprotein B-neutralizing antibody structure at 2.8 Å uncovers a critical domain for herpesvirus fusion initiation. *Nat Commun* **11**, 4141, doi:10.1038/s41467-020-17911-0 (2020).
- 5 Oliver, S. L. *et al.* The N-terminus of varicella-zoster virus glycoprotein B has a functional role in fusion. *PLoS Pathog* **17**, e1008961, doi:10.1371/journal.ppat.1008961 (2021).
- 6 Vanarsdall, A. L., Howard, P. W., Wisner, T. W. & Johnson, D. C. Human Cytomegalovirus gH/gL Forms a Stable Complex with the Fusion Protein gB in Virions. *PLoS Pathog* **12**, e1005564, doi:10.1371/journal.ppat.1005564 (2016).
- 7 Si, Z. *et al.* Different functional states of fusion protein gB revealed on human cytomegalovirus by cryo electron tomography with Volta phase plate. *PLoS Pathog* **14**, e1007452, doi:10.1371/journal.ppat.1007452 (2018).
- 8 Pataki, Z., Rebolledo Viveros, A. & Heldwein, E. E. Herpes Simplex Virus 1 Entry Glycoproteins Form Complexes before and during Membrane Fusion. *mBio*, e0203922, doi:10.1128/mbio.02039-22 (2022).
- 9 Rogalin, H. B. & Heldwein, E. E. Interplay between the Herpes Simplex Virus 1 gB Cytodomain and the gH Cytotail during Cell-Cell Fusion. *J Virol* **89**, 12262-12272, doi:10.1128/JVI.02391-15 (2015).
- 10 Pataki, Z., Sanders, E. K. & Heldwein, E. E. A surface pocket in the cytoplasmic domain of the herpes simplex virus fusogen gB controls membrane fusion. *PLoS Pathog* **18**, e1010435, doi:10.1371/journal.ppat.1010435 (2022).
- 11 Yang, E., Arvin, A. M. & Oliver, S. L. The cytoplasmic domain of varicella-zoster virus glycoprotein H regulates syncytia formation and skin pathogenesis. *PLoS Pathog* **10**, e1004173, doi:10.1371/journal.ppat.1004173 (2014).

- 12 Pak, M. A. *et al.* Using AlphaFold to predict the impact of single mutations on protein stability and function. *PLoS One* **18**, e0282689, doi:10.1371/journal.pone.0282689 (2023).

Reviewers' Comments:

Reviewer #1:

Remarks to the Author:

This work represents a significant contribution to the understanding of the mechanism of herpesvirus entry into cells. The authors thoughtfully addressed each of my original concerns and made helpful edits to the manuscript. The responses to the other reviews are appropriate. The addition of the Molecular Dynamics to the movie add substantially to the paper.

Reviewer #3:

Remarks to the Author:

I appreciate the analysis the authors have provided in the form of the molecular dynamics flexible fitting to dock a homology model for their gB (based upon another herpesvirus gB structure). Also, the analysis regarding potential missing wedge effects on the gB classes is encouraging and helps rule out positional effects as a source for the different structural classes they report. The authors have done a reasonable job of responding to my original comments, however, a main concern I still have is still that there is no link between the observed detailed conformations seen with the proline mutant and any functional states. And thus it remains unclear how the observed conformational classes relate to the process of membrane fusion.

Also, in the present study there is no dynamic or energetic information that can be extracted from the observation of multiple conformers in a single or a couple samples. The authors seemed to conflate conformational heterogeneity and dynamics. Intrinsic conformational heterogeneity could result from numerous other factors such as local environment on virus surface, differences in glycosylation, disulfide bonding or proline isomerization for example. Thus inference about kinetic trapping, energetics and dynamics based upon the cryo-EM data in the absence of other solution-phase experiments that probe conformational movements are rather speculative and in my view better to be avoided or strictly reserved for the discussion where it is made clear that the data don't directly inform on dynamics and energetics.

A few other points would be helpful to reconsider in my view:

- 1) I still believe the alpha fold part is not very informative and potentially misleading and isn't needed for this study.
- 2) The potential for inter-spike contacts to influence the conformation of gB In the membrane-presented system that displays high densities of gB has not been addressed as a potential source of the conformational perturbations.
- 3) The role of the DIII mutations themselves in giving rise to the classes of gB conformation. Could isomerization of the introduced proline be related to class 1 and 2?
- 4) It has not been demonstrated that these alternate conformations are "transition states" related to gB's fusion activity. Thus the title should be modified to moderate this claim.

Specific item:

If: "mAb 93k very likely binds to both prefusion and postfusion VZV gB," how can it be used to: "test for the preservation of its conformational epitope in DIV of gB, supporting the prefusion nature of this mutant protein."

...and also: "93k mAb binding to gB[H527P] was comparable to WT gB (Fig. 3c), indicating

preservation of the conformational 93k epitope and confirming that the gB[H527P] cryo-EM structures represent forms of gB locked in prefusion conformations.”

Since WT is much more prone to adopt the post-fusion state, the similar reactivity of 93k to WT and H527P would seem to reduce the certainty that this antibody binding supports that the gB form is indeed the prefusion conformation.

In their rebuttal, the authors state:

“Recent work by Pataki et al. showed that the proline stabilized, fusion incompetent, prefusion HSV-1 gB[H516P] could bind gH-gL at similar levels compared to WT gB 8, demonstrating that locked gB molecules can interact with gH-gL. Together, these data raise questions about the role of gH-gL in fusion activation rather than the nature of the kinetically locked prefusion structures adopted by VZV gB[H527P].”

I am not clear on this rationale. Receptor or other environmentally induced changes to gH-gL could be communicated to gB leading to activation, just because they can also interact w a conformationally locked form of gB does not negate their role in activation. Or perhaps they are released from that interaction with a prefusion gB form. Are these not possible/plausible scenarios, or perhaps I missed something?

Reviewer #1 (Remarks to the Author):

This work represents a significant contribution to the understanding of the mechanism of herpesvirus entry into cells. The authors thoughtfully addressed each of my original concerns and made helpful edits to the manuscript. The responses to the other reviews are appropriate. The addition of the Molecular Dynamics to the movie add substantially to the paper.

Author reply: Thank you! We appreciated this positive evaluation of our revised manuscript and response to the reviewers' comments.

Reviewer #3 (Remarks to the Author):

I appreciate the analysis the authors have provided in the form of the molecular dynamics flexible fitting to dock a homology model for their gB (based upon another herpesvirus gB structure). Also, the analysis regarding potential missing wedge effects on the gB classes is encouraging and helps rule out positional effects as a source for the different structural classes they report. The authors have done a reasonable job of responding to my original comments, however, a main concern I still have is still that there is no link between the observed detailed conformations seen with the proline mutant and any functional states. And thus it remains unclear how the observed conformational classes relate to the process of membrane fusion.

Author Reply: Thank you very much for giving us an encouraging assessment of our revision to the original manuscript and also our response to your comments. We understand that further clarification is needed to address your main concern. The key concept here is to evaluate our data together with the previously published work which demonstrates that the HSV-1 gB[H516P] mutant is locked in a prefusion conformation on extracellular vesicles but when the CTD is removed the mutant transitions fully to a postfusion conformation (Vollmer et al., Fig. S2) ¹. Thus, HSV-1 gB[H516P], and owing to the similarities, VZV gB[H527P], are unable to traverse the high free energy barrier, and these mutants are kinetically trapped in the context of lipid membranes. Our subtomogram averaging and classification goes further by revealing that VZV gB[H527P] has a minimum of two conformations in thermodynamic equilibrium that are kinetically trapped. We have provided a new figure in the supplementary information to demonstrate where we propose the intermediates fit into the grand scheme of the gB driven membrane fusion reaction (Fig. S7 in Supplemental information and also shown below to aid reading more easily).

The current model for herpesvirus membrane fusion is that the conformational change of gB from the metastable prefusion to the energetically more stable postfusion provides the effective activation energy (E_a) that drives membrane fusion. Structural studies in the field have revealed two endpoint structures but how prefusion gB transitions to postfusion gB remains a mystery in the absence of intermediate structures. All models of the intermediates proposed to be necessary for this process presented in the literature are speculation based on the knowledge gleaned from type I fusogens or the structurally related type III fusogen, VSV G. These suggest a series of major intermediates, including an extension of gB into the opposing membrane. We propose that these intermediates can be coarsely divided into three phases, early, peak, and late (Fig. S7). Based on our extensive biological data and previous studies, we are proposing that the class I and class II conformations of VZV gB[H527P] are in thermodynamic equilibrium but kinetically

trapped as very early intermediates unable to traverse the high free energy barrier imposed by the proline substitution needed to climb the “hill” toward the Ea.

Our cell-cell fusion assay demonstrated that the gB[H527P] was fusion incompetent, and the overall structures of the two classes of gB[H527P] resembled the condensed prefusion conformation reported for both HSV-1 and HCMV gB^{1,2}, suggesting that VZV gB[H527P] were caught either at prefusion-like state, or at a very early intermediate phase (Fig. S7). Additionally, it is important to recall that HSV-1 gB[H516P] readily snaps back to a postfusion conformation when only the ectodomain is expressed, indicating that the proline increases the free energy barrier but does not fully lock the structure; as soon as the constraint from CTD or lipid bilayer is removed, the protein is capable of fully transitioning to a postfusion conformation. Due to the conserved nature of herpesvirus gB, we expect VZV gB[H527P] to have the same properties, suggesting that the two classes of gB[H527P] will be capable of conformational transition in the absence of the CTD.

Figure S7. The proposed energy landscape of gB conformations for membrane fusion. The height of the free energy barrier corresponds to the effective activation energy (E_a) of the overall fusion process. The two endpoint states (prefusion and postfusion) of gB are depicted at the extremes of the curve; VZV prefusion gB based on homology modelling of HSV-1 prefusion gB[H516P]¹, and the 2.8Å VZV postfusion gB³. Theoretical fusion intermediates are indicated by three phases, early (yellow), peak (pink), and late (blue). The two classes of VZV gB[H527P] reported in the current manuscript are proposed to be arrested at either a prefusion-like state or very early fusion intermediate states (highlighted by the green oval). The relative energies for each entity are drawn for illustration purposes only.

Also, in the present study there is no dynamic or energetic information that can be extracted from the observation of multiple conformers in a single or a couple samples. The authors seemed to conflate

conformational heterogeneity and dynamics. Intrinsic conformational heterogeneity could result from numerous other factors such as local environment on virus surface, differences in glycosylation, disulfide bonding or proline isomerization for example. Thus inference about kinetic trapping, energetics and dynamics based upon the cryo-EM data in the absence of other solution-phase experiments that probe conformational movements are rather speculative and in my view better to be avoided or strictly reserved for the discussion where it is made clear that the data don't directly inform on dynamics and energetics.

Author Reply: Thank you for the further insight about conformational heterogeneity. We had removed any mention of inference to kinetic trapping from the results section in our previously revised manuscript and limited it to the discussion. We are in strong agreement that solution-phase experiments would be an exciting advance in the herpesvirus field. Unfortunately, solution-phase experiments would require isolation of the herpesvirus fusion machinery in a liposome-based system, which has not been achieved to date and beyond the scope of this manuscript. Although there are additional theoretical possibilities for the conformational heterogeneity, differences in glycosylation and disulfide bond formation can be ruled out. The glycosylation status of gB[H527P] compared to WT gB was unaffected as shown in Fig. 5B of the original manuscript. In addition, as all of the gB disulfide bonds are intra-domain and not inter-domain or inter-chain of the gB trimer, it is highly unlikely that this would have influenced the structures observed for the two classes. The comment about proline isomerization is intriguing and again beyond the scope of this manuscript. To address this hypothesis, near atomic resolution cryo-EM would be needed to determine whether proline isomerization was the driving force of the class I and II conformations; an exciting project for the future. Given the stability of the gB DIII alpha helix and the propensity of proline to predominantly favor a *trans* conformation, we predict that proline isomerization does not have a structural influence leading to the two gB[H527P] classes.

A few other points would be helpful to reconsider in my view:

1) I still believe the alpha fold part is not very informative and potentially misleading and isn't needed for this study.

Author Reply: We respectively disagree that the AlphaFold2 section is not informative. AlphaFold2 predicted the dramatic distortion in the DIII central helix for VZV gB[V528P]. This predictive structural information was very useful given that it was not feasible to obtain structural information about gB[V528P] experimentally due to an absence of extracellular vesicles produced with this mutant. The AlphaFold2 model further supported our findings where gB[V528P] was deficient at intracellular trafficking, absent from the cell surface, not properly proteolytically cleaved, and was retained in the ER likely due to the drastic misfolding in the central helix as predicted by AlphaFold2. Thus, the AlphaFold2 predictions form a valuable section of our manuscript and can infer similar biological properties to other human herpesviruses gB (Fig. S6B).

2) The potential for inter-spike contacts to influence the conformation of gB In the membrane-presented system that displays high densities of gB has not been addressed as a potential source of the conformational perturbations.

Author Reply: Thanks for raising this concept. This was addressed in our original responses to reviewer #1 and #3 by the addition of Fig. S2G, which shows the random distribution of the particles on the surface of the extracellular vesicles. Given the random nature of the distribution of the particles used to generate the cryo-ET maps, the high densities of gB as a source of conformational perturbations was deemed unlikely.

3) The role of the D111 mutations themselves in giving rise to the classes of gB conformation. Could isomerization of the introduced proline be related to class 1 and 2?

Author Reply: Although intriguing, this is beyond the scope of this manuscript. Please see our earlier response to this suggestion.

4) It has not been demonstrated that these alternate conformations are “transition states” related to gB’s fusion activity. Thus the title should be modified to moderate this claim.

Author Reply: Thank you for raising this point. Respectfully, as the title was supported by reviewer #1’s positive review, we have not edited the manuscript title. Also, as discussed in our earlier response, and reinforced by our biological findings, the two classes of VZV gB[H527P] were captured during vitrification in thermodynamic equilibrium. Currently, the most plausible explanation for the conformational heterogeneity and the inability of this mutant to fully transition to postfusion gB is because they are kinetically trapped in the context of lipid membranes. Thus, we favor the current title because it reflects the key discovery of such early transition states being stabilized by H527P mutation.

Specific item:

If: “mAb 93k very likely binds to both prefusion and postfusion VZV gB,” how can it be used to: “test for the preservation of its conformational epitope in DIV of gB, supporting the prefusion nature of this mutant protein.”

...and also: “93k mAb binding to gB[H527P] was comparable to WT gB (Fig. 3c), indicating preservation of the conformational 93k epitope and confirming that the gB[H527P] cryo-EM structures represent forms of gB locked in prefusion conformations.”

Since WT is much more prone to adopt the post-fusion state, the similar reactivity of 93k to WT and H527P would seem to reduce the certainty that this antibody binding supports that the gB form is indeed the prefusion conformation.

Author Reply: This concept was covered in our previous response to the original reviewer comments. Again, the 93k binding data should not be taken in isolation but also in the context of our previous studies^{3,4}. To clarify, the 93k epitope must be accessible on prefusion gB as it not only neutralizes VZV particles but also inhibits fusion in our gB/gH-gL-based virus-free cell-cell fusion assay. As gB[H527P] is locked in a prefusion conformation similar to WT HSV-1 and HCMV gB, we are confident that not only does 93k bind to prefusion VZV gB but that it does so at comparable levels to postfusion gB. We provided a new supplementary figure (Fig. S3B) in our previously revised manuscript to demonstrate that the 93k epitope was accessible for class I and class II gB[H527P], further supporting our original conclusions.

In their rebuttal, the authors state:

“Recent work by Pataki et al. showed that the proline stabilized, fusion incompetent, prefusion HSV-1 gB[H516P] could bind gH-gL at similar levels compared to WT gB 8, demonstrating that locked gB molecules can interact with gH-gL. Together, these data raise questions about the role of gH-gL in fusion activation rather than the nature of the kinetically locked prefusion structures adopted by VZV gB[H527P].”

I am not clear on this rationale. Receptor or other environmentally induced changes to gH-gL could be communicated to gB leading to activation, just because they can also interact w a conformationally locked form of gB does not negate their role in activation. Or perhaps they are released from that interaction with a prefusion gB form. Are these not possible/plausible scenarios, or perhaps I missed something?

Author Reply: The above statement quoted by the reviewer was from our response to the original reviewer comments...

“It is generally believed that the additional gH and gL VZV glycoproteins play a key role in modulating gB activity, thus in the absence of these key proteins, it seems difficult to draw definitive conclusions about the nature of gB’s prefusion conformation as it would exist on virions, let alone as relates to fusion intermediates.”

We apologize for any potential confusion. However, in our original response to the reviewers, we used the above rationale in the broader context to support the prefusion nature of VZV gB[H527P] in the absence of gH and gL because even in the extracellular vesicle system used in this and previous studies, both HSV-1 and VZV WT gB transition to postfusion in the absence of gH-gL. Critically, the gH/gL homodimer is not required to stabilize WT HSV-1 gB because, when expressed alone, 70% of the gB trimers were identified in a prefusion conformation on extracellular vesicles^{1,5}. In addition, the proline substitution at H527 for VZV gB prevented such a transition, which was not due to the lack of gH-gL, as confirmed by the inactivation of the gB[H527P] mutant virus in our BAC mutagenesis studies shown in this manuscript. Pataki et al. simply showed that gH-gL could interact with the locked HSV-1 gB[H516P] to a comparable level as WT gB. Given that their study does not distinguish the binding of gH-gL to WT gB in its prefusion or postfusion conformation, we think these data raise questions about the function of gH-gL in fusion activation because, presumably, gH-gL should disassociate from WT gB after its transition to a postfusion conformation.

References

- 1 Vollmer, B. *et al.* The prefusion structure of herpes simplex virus glycoprotein B. *Sci Adv* **6**, doi:10.1126/sciadv.abc1726 (2020).
- 2 Liu, Y. *et al.* Prefusion structure of human cytomegalovirus glycoprotein B and structural basis for membrane fusion. *Sci Adv* **7**, doi:10.1126/sciadv.abf3178 (2021).
- 3 Oliver, S. L. *et al.* A glycoprotein B-neutralizing antibody structure at 2.8 Å uncovers a critical domain for herpesvirus fusion initiation. *Nat Commun* **11**, 4141, doi:10.1038/s41467-020-17911-0 (2020).

- 4 Oliver, S. L. *et al.* The N-terminus of varicella-zoster virus glycoprotein B has a functional role in fusion. *PLoS Pathog* **17**, e1008961, doi:10.1371/journal.ppat.1008961 (2021).
- 5 Zeev-Ben-Mordehai, T. *et al.* Two distinct trimeric conformations of natively membrane-anchored full-length herpes simplex virus 1 glycoprotein B. *Proc Natl Acad Sci U S A* **113**, 4176-4181, doi:10.1073/pnas.1523234113 (2016).

Reviewers' Comments:

Reviewer #3:

Remarks to the Author:

I stand by my previous points, but believe this is notable work that can contribute new data and insights to the field.